# Hierarchical Gaussian Mixture Normalizing Flows Modeling for Multi-Class Anomaly Detection

## Abstract

Multi-class anomaly detection (AD) is one of the most challenges for anomaly detection. For such a challenging task, popular normalizing flow (NF) based AD methods may fall into a "homogeneous mapping" issue, where the NF-based AD models are biased to generate similar latent representations for both normal and abnormal features, and thereby lead to a high missing rate of anomalies. In this paper, we propose a novel **H**ierarchical **G**aussian mixture normalizing flows modeling method for accomplishing multi-class **A**nomaly **D**etection, which we call HGAD. Our HGAD consists of two key components: inter-class Gaussian mixture prior modeling and intra-class mixed class centers learning. Compared to the previous NF-based AD methods, the hierarchical Gaussian mixture modeling approach can bring stronger representation capability to the latent space of normalizing flows, so that even complex multi-class distribution can be well represented and learned in the latent space. In this way, we can avoid mapping different class distributions into the same single Gaussian prior, thus effectively avoiding or mitigating the "homogeneous mapping" issue. We further find that the more distinguishable different class centers, the more conducive to avoiding the bias issue. Thus, we further propose a mutual information maximization loss for better structuring the latent feature space. We evaluate our method on four real-world AD benchmarks, where we can significantly improve the previous NF-based AD methods and also outperform the SOTA unified AD methods. Code will be available online.

## 1 Introduction

Anomaly detection has received increasingly wide attentions and applications in different scenarios, such as industrial defect detection (Bergmann et al., 2019; Roth et al., 2022), video surveillance (Acsintoae et al., 2022; Sultani et al., 2018), medical lesion detection (Tian et al., 2021; Zhang et al., 2021), and road anomaly detection (Vojir et al., 2021; Biase et al., 2021). Considering the highly scarce anomalies and diverse normal classes, most previous AD studies have mainly devoted to unsupervised one-class learning, *i.e.*, learning one specific AD model by only utilizing one-class normal samples and then detecting anomalies in this class. However, such a one-for-one paradigm would require more human labor, time, and computation costs when training and testing many product categories, and also underperform when the one normal class has large intra-class diversity.

In this work, considering the unified AD ability, we aim to tackle a more practical task: multi-class anomaly detection. As shown in Fig. 1b, one unified model is trained with normal samples from multiple classes, and the objective is to detect anomalies for all these classes without any fine-tuning. Nonetheless, solving such a task is quite challenging. Currently, there are two reconstruction-based AD methods for tackling the challenging multi-class AD task, UniAD (You et al., 2022) and PMAD (Yao et al., 2023b). But the reconstruction-based methods may fall into the "identical shortcut reconstruction" dilemma (You et al., 2022), where anomalies can also be well reconstructed, resulting in the failure of anomaly detection. UniAD and PMAD attempt to mask the adjacent or suspicious anomalies to avoid identical reconstruction. However, due to the diverse scale and shape of anomalies, the masking mechanism cannot completely avoid the abnormal information leakage during reconstruction, the risk of identical reconstruction is still existing. To this end, we consider designing unified AD model from the normal data distribution learning perspective. The advantage is

that we will no longer face the abnormal information leakage risk in principle, as there is no need to reconstruct anomalies as normals for detecting anomalies. Specifically, we employ normalizing flows (NF) to learn the normal data distribution (Gudovskiy et al., 2022).

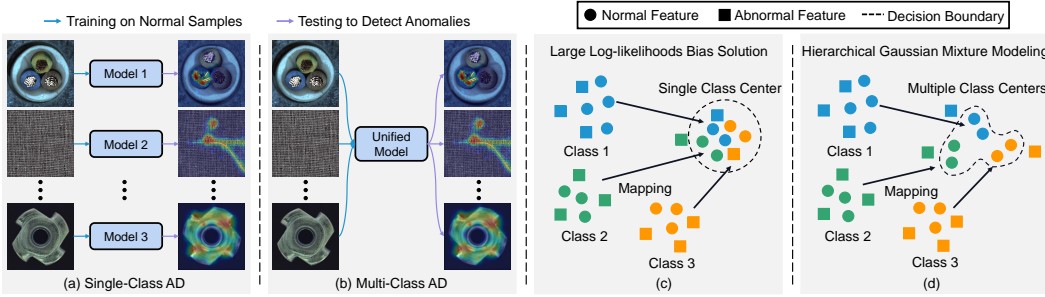

Figure 1: **Anomaly detection task settings**. We aim to implement one unified AD model (b). (c) Mapping all input features to the same latent class center may induce the risk of learning a biased solution. (d) We propose a hierarchical Gaussian mixture normalizing flows modeling method for more effectively capturing the complex multi-class distribution.

However, we find that the NF-based AD methods perform unsatisfactorily when applied to the multi-class AD task. They usually fall into a "homogeneous mapping" issue (see Sec. 3.2), where the NF-based AD models are biased to generate large log-likelihoods for both normal and abnormal inputs (see Fig. 2b). We further explain this issue as: The multi-class distribution is far more diverse and usually multi-modal. However, conventional NF-based AD methods (Gudovskiy et al., 2022; Yu et al., 2021) employ the uni-modal Gaussian prior to learn the invertible mapping. This can be seen as learning a mapping from a heterogeneous space to the latent homogeneous space. To learn the mapping well, the network may be prompted to take a bias to concentrate on the coarse-grained common characteristics (*e.g.*, local pixel correlations) and suppress the fine-grained distinguishable characteristics (*e.g.*, semantic content) among different class features (Kirichenko et al., 2020). Consequently, the network homogeneously maps different class features to the close latent embeddings. Thus, even anomalies can obtain large log-likelihoods and become less distinguishable.

To address this issue, we first empirically confirm that mapping to multiple latent class centers is effective to prevent the model from learning the bias (see Fig. 2c). Accordingly, we propose to model NF-based AD networks with inter-class Gaussian mixture prior for more effectively capturing the complex multi-class distribution. Second, we argue that the inter-class Gaussian mixture prior can only ensure the features are drawn to the whole prior distribution but lacks inter-class repulsion, still resulting in a much weaker discriminative ability for different class features. This may cause different class centers to collapse into the same center. To further increase the inter-class discriminability, we propose a mutual information maximization loss to introduce the class repulsion property to the model for better structuring the latent feature space, where the class centers can be pushed away from each other. Third, we introduce an intra-class mixed class centers learning strategy that can urge the model to learn diverse normal patterns even within one class. Finally, we form a hierarchical Gaussian mixture normalizing flows modeling approach for multi-class anomaly detection, which we call HGAD. Our method can dramatically improve the unified AD performance of the previous single-class NF-based AD methods (*e.g*, CFLOW-AD), boosting the AUROC from 89.0%/94.0% to 98.4%/97.9%, and also outperform the SOTA unified AD methods (*e.g*, UniAD).

## 2 RELATED WORK

**Anomaly Detection.** 1) *Reconstruction-based approaches* are the most popular AD methods. These methods rely on the assumption that models trained by normal samples would fail in abnormal regions. Many previous works attempt to train AutoEncoders (Park et al., 2020; Zavrtanik et al., 2021), Variational AutoEncoders (Liu et al., 2020) and GANs (Schlegl et al., 2017; Akcay et al., 2018) to reconstruct the input images. However, these methods face the "identical shortcut" problem (You et al., 2022). 2) *Embedding-based approaches* recently show better AD performance by using ImageNet pre-trained networks as feature extractors (Bergman et al., 2020; Cohen & Hoshen, 2020).

PaDiM (Defard et al., 2021) extract pre-trained features to model Multivariate Gaussian distribution for normal samples, then utilize Mahalanobis distance to measure the anomaly scores. PatchCore (Roth et al., 2022) extends on this line by utilizing locally aggregated features and introducing greedy coreset subsampling to form nominal feature banks. 3) *Knowledge distillation* assumes that the student trained to learn the teacher on normal samples could only regress normal features but fail in abnormal features (Bergmann et al., 2020). Recent works mainly focus on feature pyramid Salehi et al. (2021); Wang et al. (2021), reverse distillation (Deng & Li, 2022), and asymmetric distillation (Rudolph et al., 2023). 4) *Unified AD approaches* attempt to train a unified AD model to accomplish anomaly detection for multiple classes. UniAD (You et al., 2022), PMAD (Yao et al., 2023b) and OmniAL (Zhao, 2023) are three existing methods in this new direction. UniAD is a transformer-based reconstruction model with three improvements, it can perform well under the unified case by addressing the "identical shortcut" issue. PMAD is a MAE-based patch-level reconstruction model, which can learn a contextual inference relationship within one image rather than the class-dependent reconstruction mode. OmniAL is a unified CNN framework with anomaly synthesis, reconstruction and localization improvements. To prevent the identical reconstruction, OmniAL trains the model with proposed panel-guided synthetic anomaly data rather than directly using normal data.

**Normalizing Flows in Anomaly Detection.** In anomaly detection, normalizing flows are employed to learn the normal data distribution (Rudolph et al., 2021; Gudovskiy et al., 2022; Yu et al., 2021; Yao et al., 2023a), which maximize the log-likelihoods of normal samples during training. Rudolph *et al.* (Rudolph et al., 2021) first employ NFs for anomaly detection by estimating the distribution of pre-trained features. In CFLOW-AD (Gudovskiy et al., 2022), the authors further construct NFs on multi-scale feature maps to achieve anomaly localization. Recently, fully convolutional normalizing flows (Rudolph et al., 2022; Yu et al., 2021) have been proposed to improve the accuracy and efficiency of anomaly detection. In BGAD (Yao et al., 2023a), the authors propose a NF-based AD model to tackle the supervised AD task. In this paper, we mainly propose a novel NF-based AD model (HGAD) with three improvements to achieve much better unified AD performance.

## 3 METHOD

### 3.1 PRELIMINARY OF NORMALIZING FLOW BASED ANOMALY DETECTION

In subsequent sections, upper case letters denote random variables (RVs) (*e.g.*, $X$) and lower case letters denote their instances (*e.g.*, $x$). The probability density function of a RV is written as $p(X)$, and the probability value for one instance as $p_X(x)$. The normalizing flow models (Dinh et al., 2017; Kingma & Dhariwal, 2019) can fit an arbitrary distribution $p(X)$ by a tractable latent base distribution with $p(Z)$ density and a bijective invertible mapping $\varphi : X \in \mathbb{R}^d \to Z \in \mathbb{R}^d$. Then, according to the change of variable formula (Villani, 2003), the log-likelihood of any $x \in X$ can be estimated as:

$$\log p_\theta(x) = \log p_Z(\varphi_\theta(x)) + \log|\det J| \tag{1}$$

where $\theta$ means the learnable model parameters, and we use $p_\theta(x)$ to denote the estimated probability value of feature $x$ by the model $\varphi_\theta$. The $J = \bigtriangledown_x \varphi_\theta(x)$ is the Jacobian matrix of the bijective transformation ($z = \varphi_\theta(x)$ and $x = \varphi_\theta^{-1}(z)$). The model parameters $\theta$ can be optimized by maximizing the log-likelihoods across the training distribution $p(X)$. The loss function is defined as:

$$\mathcal{L}_m = \mathbb{E}_{x \sim p(X)}[-\log p_\theta(x)] \tag{2}$$

In anomaly detection, the latent variables $Z$ for normal features are usually assumed to obey $\mathcal{N}(0, \mathbb{I})$ for simplicity (Rudolph et al., 2021). By replacing $p_Z(z) = (2\pi)^{-\frac{d}{2}} e^{-\frac{1}{2}z^T z}$, $z = \varphi_\theta(x)$ in Eq. 1, the loss function in Eq. 2 can be written as:

$$\mathcal{L}_m = \mathbb{E}_{x \sim p(X)}\left[\frac{d}{2}\log(2\pi) + \frac{1}{2}\varphi_\theta(x)^T \varphi_\theta(x) - \log|\det J|\right] \tag{3}$$

After training, the log-likelihoods of the input features can be exactly estimated by the trained normalizing flow models as $\log p_\theta(x) = -\frac{d}{2}\log(2\pi) - \frac{1}{2}\varphi_\theta(x)^T \varphi_\theta(x) + \log|\det J|$. Next, we can convert log-likelihoods to probabilities via exponential function: $p_\theta(x) = e^{\log p_\theta(x)}$. As we maximize log-likelihoods for normal features in Eq. 2, the estimated probabilities $p_\theta(x)$ can directly measure the normality. Thus, the anomaly scores can be calculated by $s(x) = 1 - p_\theta(x)$.

## 3.2 Revisiting Normalizing Flow Based Anomaly Detection Methods

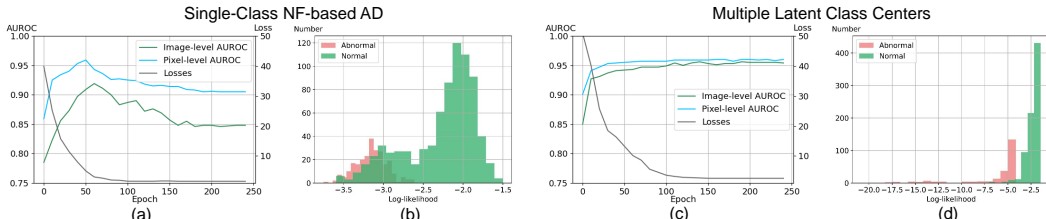

Figure 2: **Comparison between single-class and multi-class NF-based AD methods on MVTecAD**. (a) and (c) show the training losses and the testing anomaly detection and localization AUROCs. (b) shows that the single-class NF-based AD model may have an obvious norma-abnormal overlap, while ours (d) can bring better normal-abnormal distinguishability.

Under the multi-class AD task, we follow the NF-based anomaly detection paradigm (Rudolph et al., 2021; Gudovskiy et al., 2022) and reproduce the FastFlow (Yu et al., 2021) to estimate log-likelihoods of the features extracted by a pre-trained backbone. We then convert the estimated log-likelihoods to anomaly scores and evaluate the AUROC metric every 10 epochs. As shown in Fig. 2a, after a period of training, the performance of the model drops severely while the losses continue going extremely small. Accordingly, the overall log-likelihoods become much large. We attribute this phenomenon to the "homogeneous mapping" issue, where the normalizing flows may map all inputs to much close latent variables and then present large log-likelihoods for both normal and abnormal features, thus failing to detect anomalies. This speculation is empirically verified by the visualization results in Fig. 2b (more results in App. Fig. 5), where the normal and abnormal log-likelihoods are highly overlapped. As we explained in Sec. 1, the phenomenon may come from that the model excessively suppresses the fine-grained distinguishable characteristics between normal and abnormal features. However, as shown in Fig. 2c and 2d, when using multiple latent class centers and mapping different class features to their corresponding class centers, the model can more effectively avoid highly overlapped normal and abnormal log-likelihoods, indicating a slighter log-likelihoods bias problem. This encourages us to analyze as follows.

Below, we denote normal features as $x_n \in \mathbb{R}^d$ and abnormal features as $x_a \in \mathbb{R}^d$, where $d$ is the channel dimension. We provide a rough analysis using a simple one coupling layer normalizing flow model. When training, the forward affine coupling (Dinh et al., 2017) can be calculated as:

$$x_1, x_2 = \text{split}(x_n) \quad \text{and} \quad z_1 = x_1; z_2 = x_2 \odot \exp(s(x_1)) + t(x_1) \quad \text{and} \quad z = \text{cat}(z_1, z_2) \quad (4)$$

where $\text{split}$ and $\text{cat}$ mean split and concatenate the feature maps along the channel dimension, $s(x_1)$ and $t(x_1)$ are transformation coefficients predicted by a learnable neural network (Dinh et al., 2017). With the maximum likelihood loss in Eq. 3 pushing all $z$ to fit $\mathcal{N}(0, \mathbb{I})$, the model has no need to distinguish different class features. Thus, it is more likely to take a bias to predict all $s(\cdot)$ to be very small negative numbers ($\rightarrow -\infty$) and $t(\cdot)$ close to zero. The impact is that the model could also fit $x_a$ to $\mathcal{N}(0, \mathbb{I})$ well with the bias, failing to detect anomalies. However, if we use multiple latent class centers and map different class features to their corresponding class centers, the model is harder to simply take a bias solution. Instead, $s(\cdot)$ and $t(\cdot)$ must be highly related to input features. Considering that $s(\cdot)$ and $t(\cdot)$ in the trained model are relevant to normal features, the model thus could not fit $x_a$ well. We think that the above rough analysis can also be applied to multiple layers. Because the output of one coupling layer will tend to 0 when $s(\cdot)$ and $t(\cdot)$ of the layer are biased. From Eq. 4, we can see that when the output of one coupling layer is close to 0, the output of the next layer will also tend to 0. Therefore, the output after multiple layers will tend to 0, the network is still biased.

## 3.3 Hierarchical Gaussian Mixture Normalizing Flows Modeling

**Overview.** As shown in Fig. 3, our HGAD is composed of a feature extractor, a normalizing flow model (details in App. C), and the hierarchical Gaussian mixture modeling. First, the features extracted by a fixed pre-trained backbone are sent into the normalizing flow model to transform into the latent embeddings. Then, the latent embeddings are used to fit the hierarchical Gaussian mixture

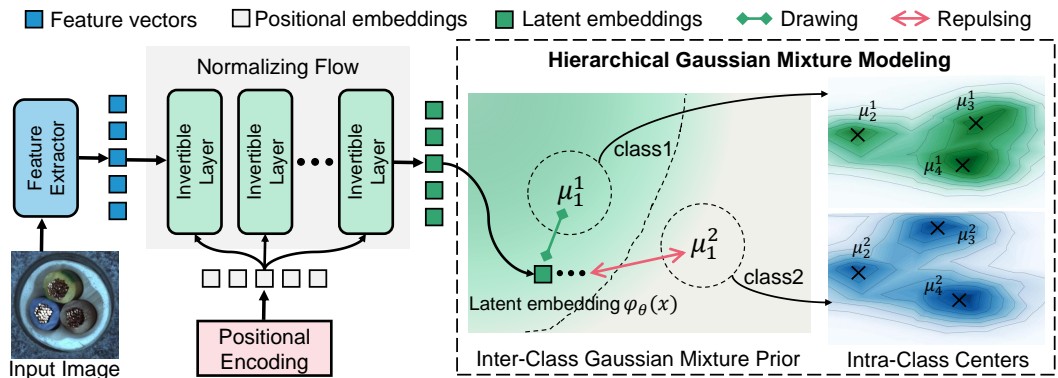

Figure 3: **Model overview**. The extracted feature vectors are sent into the normalizing flow model for transforming into latent embeddings. Positional embeddings are added to each invertible layer Gudovskiy et al. (2022). The latent embeddings are used to fit the hierarchical Gaussian mixture prior, which can assist the model against learning the "homogeneous mapping".

prior during training. The pseudo-code of our HGAD is provided in Alg. 1. In the subsequent sections, we will describe the hierarchical Gaussian mixture modeling and the specific losses in detail.

---

**Algorithm 1** HGAD: **H**ierarchical **G**aussian mixture modeling for multi-class **A**nomaly **D**etection

---

**Input:** Input image $I \in \mathbb{R}^{H \times W \times 3}$, Class label $y \in \{1, \dots, Y\}$
1: Initialization: Class centers $\mu_y \leftarrow \boldsymbol{y}, y \in \{1, \dots, Y\}$, Learnable vector $\psi \leftarrow \boldsymbol{0}$, Total loss $\mathcal{L}_{all} \leftarrow 0$
2: Feature Extraction: we extract $K$ feature maps, denoted as $X^k, k \in \{1, \dots, K\}$
3: **for** each feature $x \in X^k$ **do**
4:      Obtain the latent representation: $\varphi_\theta(x)$
5:      Calculate all logarithmic class weights: $c_y = \text{logsoftmax}_y(\psi), y \in \{1, \dots, Y\}$
6:      Calculate **inter-class loss**: $\mathcal{L}_g$ based on Eq. 6
7:      Calculate **mutual information maximization loss**: $\mathcal{L}_{mi}$ based on Eq. 8
8:      Calculate **entropy loss**: $\mathcal{L}_e$ based on Eq. 9
9:      Calculate **intra-class loss**: $\mathcal{L}_{in}$ based on Eq. 10
10:     Calculate the overall loss: $\mathcal{L} = \lambda_1 \mathcal{L}_g + \lambda_2 \mathcal{L}_{mi} + \mathcal{L}_e + \mathcal{L}_{in}$
11:     Update $\mathcal{L}_{all} \leftarrow \mathcal{L}_{all} + \mathcal{L}$
12: **end for**
13: Update mean loss $\mathcal{L}_{mean} = \mathcal{L}_{all}/N$, $N$ is the total number of features
**Output:** Mean loss $\mathcal{L}_{mean}$

---

**Modeling Normalizing Flows with Inter-Class Gaussian mixture Prior.** As discussed in Sec. 3.2, using multiple latent class centers can suffer a slighter log-likelihoods bias problem. To this end, to further better fit the complex multi-class normal distribution in the latent space, a natural way is to extend the single-class Gaussian prior to the inter-class Gaussian mixture prior. Specifically, a Gaussian mixture model with class-dependent means $\mu_y$ and covariance matrices $\Sigma_y$, where $y$ means the class labels, is used as the prior distribution for the latent variables $Z$:

$$p(Z|y) = \mathcal{N}(\mu_y, \Sigma_y) \quad \text{and} \quad p_Z(z) = \sum_y p(y)\mathcal{N}(z; \mu_y, \Sigma_y) \qquad (5)$$

For simplicity, we also use unit matrix $\mathbb{I}$ to replace all the class-dependent covariance matrices $\Sigma_y$. To urge the network to adaptively learn the class weights, we parameterize the class weights $p(Y)$ through a learnable vector $\psi$, with $p(y) = \text{softmax}_y(\psi)$, where the subscript of the softmax operator denotes the class index of the calculated softmax value. The use of the softmax can ensure that $p(y)$ stays positive and sums to one. The $\psi$ can be initialized to 0. With the parameterized $p(Y)$, we can derive the loss function with the inter-class Gaussian mixture prior as follows (the detailed derivation

is in App. E):

$$\mathcal{L}_g = \mathbb{E}_{x \sim p(X)} \left[ -\operatorname*{logsumexp}_{y} \left( -\frac{||\varphi_\theta(x) - \mu_y||_2^2}{2} + c_y \right) - \log|\det J| + \frac{d}{2}\log(2\pi) \right] \quad (6)$$

where $c_y$ denotes logarithmic class weights and is defined as $c_y := \log p(y) = \operatorname{logsoftmax}_y(\psi)$, and the subscript $y$ of the $\operatorname{logsumexp}$ operator denotes summing the $\exp$ values of all classes.

**Mutual Information Maximization.** Next, we further argue that the inter-class Gaussian mixture prior can only ensure the latent features are drawn together to the whole prior distribution (parameterized by $\{\mu_y, \psi_y\}_{y=1}^Y$), where the $\{p(y)\}_{y=1}^Y$ can control the contribution of different class centers to the log-likelihood estimation value $\log p_\theta(x)$. This means that the loss function in Eq. 6 only has the drawing characteristic to make the latent features fit the Gaussian mixture prior, but without the repulsion property for separating among different classes, still resulting in a much weaker discriminative ability for different class features. As the class centers are randomly initialized, this may cause different class centers to collapse into the same center. To address this, we consider that the latent feature $z$ with class label $y$ should be drawn close to its corresponding class center $\mu_y$ as much as possible while far away from the other class centers. From the information theory perspective, this means that the mutual information $I(Y, Z)$ should be large enough. So, we propose a mutual information maximization loss to introduce the class repulsion property for increasing the class discrimination ability. The loss function is defined as follows (the derivation is in App. E):

$$\mathcal{L}_{mi} = -\mathbb{E}_{y \sim p(Y)}[-\log p(y)] - \mathbb{E}_{(x,y) \sim p(X,Y)} \left[ \log \frac{p(y)p(\varphi_\theta(x)|y)}{\sum_{y'} p(y')p(\varphi_\theta(x)|y')} \right] \quad (7)$$

By replacing $p(\varphi_\theta(x)|y)$ with $\mathcal{N}(\varphi_\theta(x); \mu_y, \mathbb{I})$ in Eq. 7, we can derive the following practical loss format (the detailed derivation is in App. E):

$$\mathcal{L}_{mi} = -\mathbb{E}_{(x,y) \sim p(X,Y)} \left[ \operatorname*{logsoftmax}_{y} \left( -\frac{||\varphi_\theta(x) - \mu_{y'}||_2^2}{2} + c_{y'} \right) - c_y \right] \quad (8)$$

where $\mu_{y'}$ means all the other class centers except for $\mu_y$, and the subscript $y$ of the $\operatorname{logsoftmax}$ operator denotes calculating logsoftmax value for class $y$. Note that we also use this representation way for softmax calculation in the following sections.

In addition to the mutual information maximization loss, we propose that we can also introduce the class repulsion property by minimizing the inter-class entropy. We use the $-||\varphi_\theta(x) - \mu_y||_2^2/2$ as the class logits for class $y$, and then define the entropy loss as follows (a standard entropy formula):

$$\mathcal{L}_e = \mathbb{E}_{x \sim p(X)} \left[ \sum_y -\operatorname*{softmax}_y(-||\varphi_\theta(x) - \mu_{y'}||_2^2/2) \cdot \operatorname*{logsoftmax}_y(-||\varphi_\theta(x) - \mu_{y'}||_2^2/2) \right] \quad (9)$$

**Learning Intra-Class Mixed Class Centers.** In real-world scenarios, even one object class may contain diverse normal patterns. Thus, to better model intra-class distribution, we further extend the Gaussian prior $p(Z|y) = \mathcal{N}(\mu_y, \Sigma_y)$ to mixture Gaussian prior $p(Z|y) = \sum_{i=1}^M p_i(y)\mathcal{N}(\mu_i^y, \Sigma_i^y)$, where $M$ is the number of intra-class latent centers. We can directly replace the $p(Z|y)$ in Eq. 5 and derive the corresponding loss function $\mathcal{L}_g$ in Eq. 6 (see App. Eq. 18). However, the initial latent features $Z$ usually have large distances with the intra-class centers $\{\mu_i^y\}_{i=1}^M$, this will cause the $p(z|y), z \in Z$ close to 0. After calculating the logarithm function, it is easy to cause the loss to be numerically ill-defined (NaN), making it fail to be optimized. To this end, we propose to decouple the inter-class Gaussian mixture prior fitting and the intra-class latent centers learning. This decoupling strategy is more conducive to learn class centers as we form a coarse-to-fine optimization process. Specifically, for each class $y$, we learn a main class center $\mu_1^y$ and the delta vectors $\{\Delta\mu_i^y\}_{i=1}^M$ ($\Delta\mu_1^y$ is fixed to 0), which mean the offset values from the main center and are used to represent the other intra-class centers: $\mu_i^y = \{\mu_1^y + \Delta\mu_i^y\}_{i=1}^M$. Then, we can directly employ the Eq. 6 to optimize the main center $\mu_1^y$. When learning the other intra-class centers, we detach the main center $\mu_1^y$ from the gradient graph and only optimize the delta vectors by the following loss function:

$$\mathcal{L}_{in} = \mathbb{E}_{(x,y) \sim p(X,Y)} \left[ -\operatorname*{logsumexp}_{i} \left( -\frac{||\varphi_\theta(x) - (SG[\mu_1^y] + \Delta\mu_i^y)||_2^2}{2} + c_i^y \right) - \log|\det J| \right] \quad (10)$$

where $SG[\cdot]$ means to stop gradient backpropagation, $c_i^y$ denotes logarithmic intra-class center weights and is defined as $c_i^y := \log p_i(y) = \text{logsoftmax}_i(\psi_y)$.

**Overall Loss Function.** The overall training loss function is the combination of the Eq. 6, Eq. 8, Eq. 9 and Eq. 10, as follows:

$$\mathcal{L} = \lambda_1 \mathcal{L}_g + \lambda_2 \mathcal{L}_{mi} + \mathcal{L}_e + \mathcal{L}_{in} \tag{11}$$

where the $\lambda_1$ and $\lambda_2$ are used to trade off the loss items and are set to 1 and 100 by default. In App. A, we summarize the supervision information required by our method and other methods, we also further discuss the limitations, applications, effective guarantee, and complexity of our method. We also provide an information-theoretic view in App. F.

### 3.4 ANOMALY SCORING

Formally, we define $K$ as the subset including the indexes of feature maps for use. For each test input feature $x^k$ from level-$k$, $k \in K$, we can calculate its intra-class log-likelihood $\log p_\theta(x^k) = \text{logsumexp}_i(-||\varphi_\theta(x^k) - \mu_i^y||_2^2/2 + c_i^y) + \log|\det J| - d/2\log(2\pi)$ and inter-class negative entropy $nh(x^k) = \sum_y \text{softmax}_y(-||\varphi_\theta(x^k) - \mu_1^{y'}||_2^2/2) \cdot \text{logsoftmax}_y(-||\varphi_\theta(x^k) - \mu_1^{y'}||_2^2/2)$. Note that $y$ in $c_i^y$ denotes which class $x^k$ belongs to, not whether $x^k$ is normal or abnormal. Next, we convert the log-likelihood to probability $p_\theta(x^k) = e^{\log p_\theta(x^k)}$. Then, we upsample all $p_\theta(x^k)$ in the level-$k$ to the input image resolution $(H \times W)$ using bilinear interpolation $P_k = b(p_\theta(x^k)) \in \mathbb{R}^{H \times W}$. Finally, we calculate anomaly score map $S_l$ by aggregating all upsampled probabilities as $S_l = \max(\sum_{k=1}^K P_k) - \sum_{k=1}^K P_k$. For the inter-class negative entropy, we also follow the above steps to convert to anomaly score map $S_e$. Then the final anomaly score map is obtained by combining the two maps $S = S_l \odot S_e$, where the $\odot$ is the element-wise multiplication. In this way, even if anomalies fall into the inter-class Gaussian mixture distribution, they are usually in the low-density regions among the inter-class class centers. So, we can still ensure that anomalies are out-of-distribution through intra-class log-likelihoods (please see App. A for more discussions).

## 4 EXPERIMENT

### 4.1 DATASETS AND METRICS

**Datasets.** We extensively evaluate our approach on four real-world industrial AD datasets: MVTecAD (Bergmann et al., 2019), BTAD (Mishra et al., 2021), MVTec3D-RGB (Bergmann et al., 2021), and VisA (zou et al., 2022). The detailed introduction to these datasets is provided in App. D. To more sufficiently evaluate the unified AD performance of different AD models, we combine these datasets to form a 40-class dataset, which we call Union dataset.

**Metrics.** Following prior works (Bergmann et al., 2019; 2020; Zavrtanik et al., 2021), the standard metric in anomaly detection, AUROC, is used to evaluate the performance of AD methods.

### 4.2 MAIN RESULTS

**Setup.** All the images are resized and cropped to $256 \times 256$ resolution. The feature maps from stage-1 to stage-3 of EfficientNet-b6 (Tan & Le, 2019) are used as inputs to the normalizing flow models. The parameters of the feature extractor are frozen during training. The layer numbers of the NF models are all 12. The number of inter-class centers is always equal to the number of classes in the dataset. The number of intra-class centers is set as 10 for all datasets (see ablation study in Sec. 4.3). We use the Adam (P.Kingma & Ba, 2015) optimizer with weight decay $1e^{-4}$ to train the model. The total training epochs are set as 100 and the batch size is 8 by default. The learning rate is $2e^{-4}$ initially, and dropped by 0.1 after $[48, 57, 88]$ epochs. The evaluation is run with 3 random seeds.

**Baselines.** We compare our approach with single-class AD baselines including: PaDiM (Defard et al., 2021), MKD (Salehi et al., 2021), and DRAEM (Zavrtanik et al., 2021), and SOTA unified AD methods: PMAD (Yao et al., 2023b), UniAD (You et al., 2022), and OmniAL (Zhao, 2023). We also compare with the SOTA single-class NF-based AD methods: CFLOW (Gudovskiy et al., 2022) and FastFlow (Yu et al., 2021). Under the unified case, the results of the single-class AD baselines and the NF-based AD methods are run with the publicly available implementations.

Table 1: **Anomaly detection and localization results on MVTecAD**. All methods are evaluated under the unified case. $\cdot/\cdot$ means the image-level and pixel-level AUROCs.

| Category | Baseline Methods | | | Unified Methods | | | Normalizing Flow Based Methods | | |
|---|---|---|---|---|---|---|---|---|---|
| | PaDiM | MKD | DRAEM | PMAD | UniAD | OmniAL | FastFlow | CFLOW | HGAD (Ours) |
| Carpet | 93.8/97.6 | 69.8/95.5 | 98.0/98.6 | 99.0/97.9 | 99.8/98.5 | 98.7/**99.4** | 91.6/96.7 | 98.8/97.5 | **100**±0.00/**99.4**±0.05 |
| Grid | 73.9/71.0 | 83.8/82.3 | 99.3/98.7 | 96.2/95.6 | 98.2/96.5 | **99.9/99.4** | 85.7/96.8 | 95.9/94.1 | 99.6±0.09/99.1±0.08 |
| Leather | 99.9/84.8 | 93.6/96.7 | 98.7/97.3 | **100**/99.2 | **100**/98.8 | 99.0/99.3 | 93.7/98.2 | **100**/98.1 | **100**±0.00/**99.6**±0.00 |
| Tile | 93.3/80.5 | 89.5/85.3 | 99.8/98.0 | 99.8/94.5 | 99.3/91.8 | 99.6/**99.0** | 99.2/95.8 | 97.9/92.2 | **100**±0.00/96.1±0.09 |
| Wood | 98.4/89.1 | 93.4/80.5 | **99.8**/96.0 | 99.6/89.0 | 98.6/93.2 | 93.2/**97.4** | 98.0/92.0 | 99.0/92.7 | 99.5±0.08/95.9±0.09 |
| Bottle | 97.9/96.1 | 98.7/91.8 | 97.5/87.6 | 99.8/98.4 | 99.7/98.1 | **100/99.2** | **100**/94.0 | 98.7/96.4 | **100**±0.00/98.6±0.08 |
| Cable | 70.9/81.0 | 78.2/89.3 | 57.8/71.3 | 93.5/95.4 | 95.2/**97.3** | **98.2**/97.3 | 90.9/95.2 | 80.4/92.9 | 97.3±0.26/95.2±0.49 |
| Capsule | 73.4/96.9 | 68.3/88.3 | 65.3/50.5 | 80.5/97.0 | 86.9/98.5 | 95.2/96.9 | 90.5/98.6 | 75.5/97.7 | **99.0**±0.40/**99.2**±0.05 |
| Hazelnut | 85.5/96.3 | 97.1/91.2 | 93.7/96.9 | 99.6/97.4 | 99.8/98.1 | 95.6/98.4 | 98.9/96.6 | 97.1/95.7 | 99.9±0.08/**98.8**±0.05 |
| Metal nut | 88.0/84.8 | 64.9/64.2 | 72.8/62.2 | 98.0/91.7 | 99.2/94.8 | 99.2/**99.1** | 96.5/97.2 | 87.8/84.4 | **100**±0.00/97.8±0.29 |
| Pill | 68.8/87.7 | 79.7/69.7 | 82.2/94.4 | 89.4/93.4 | 93.7/95.0 | **97.2/98.9** | 90.4/96.1 | 88.0/90.7 | 96.3±0.73/98.8±0.05 |
| Screw | 56.9/94.1 | 75.6/92.1 | 92.0/95.5 | 73.3/96.6 | 87.5/98.3 | 88.0/98.0 | 76.8/95.9 | 59.5/93.9 | **95.5**±0.16/**99.3**±0.12 |
| Toothbrush | 95.3/95.6 | 75.3/88.9 | 90.6/97.7 | 95.8/98.2 | 94.2/98.4 | **100/99.4** | 86.1/97.1 | 78.0/95.7 | 91.2±0.37/99.1±0.05 |
| Transistor | 86.6/92.3 | 73.4/71.7 | 74.8/64.5 | 97.2/93.3 | **99.8/97.9** | 93.8/93.3 | 85.7/93.8 | 86.7/92.3 | 97.7±0.21/91.9±0.26 |
| Zipper | 79.7/94.8 | 87.4/86.1 | 98.8/98.3 | 96.0/96.1 | 95.8/96.8 | **100/99.5** | 93.8/95.7 | 92.2/95.7 | **100**±0.04/99.0±0.09 |
| **Mean** | 84.2/89.5 | 81.9/84.9 | 88.1/87.2 | 94.5/95.6 | 96.5/96.8 | 97.2/**98.3** | 91.8/96.0 | 89.0/94.0 | **98.4**±0.08/97.9±0.05 |

Table 2: **Anomaly detection and localization results on BTAD, MVTec3D-RGB, VisA, and Union datasets**. All methods are evaluated under the unified case.

| Dataset | PaDiM | MKD | DRAEM | PMAD | UniAD | OmniAL | FastFlow | CFLOW | HGAD (Ours) |
|---|---|---|---|---|---|---|---|---|---|
| **BTAD** | 93.8/96.6 | 89.7/96.2 | 91.2/91.9 | 93.8/**97.3** | 94.0/97.2 | -/- | 92.9/95.3 | 93.0/96.6 | **95.2**±0.09/97.1±0.12 |
| **MVTec3D-RGB** | 77.4/96.3 | 73.5/95.9 | 73.9/95.5 | 75.4/95.3 | 77.5/**96.6** | -/- | 67.9/90.2 | 71.6/95.7 | **83.9**±0.35/**96.6**±0.05 |
| **VisA** | 86.8/97.0 | 74.2/93.9 | 85.5/90.5 | -/- | 92.8/98.1 | 87.8/96.6 | 77.2/95.1 | 88.0/95.9 | **97.1**±0.11/**98.9**±0.06 |
| **Union** | 79.0/91.4 | 72.1/88.9 | 66.4/82.7 | -/- | 86.9/95.5 | -/- | 57.2/78.8 | 55.7/82.9 | **92.3**±0.26/**96.5**±0.13 |

**Quantitative Results.** The detailed results on MVTecAD are shown in Tab. 1. We also report the results under the single-class setting in App. Tab. 6. By comparison, we can see that the performances of all baselines and SOTA single-class NF-based AD methods drop dramatically under the unified case. However, our HGAD outperforms all baselines under the unified case significantly. Compared with the single-class NF-based AD counterparts, we improve the unified AD performance from 91.8% to 98.4% and from 96.0% to 97.9%. We further indicate that our model has the same network architecture as CFLOW, and we only introduce multiple inter- and intra-class centers as extra learnable parameters. However, our method can significantly outperform CFLOW. This demonstrates that our novel designs are the keys to improving the unified AD ability of the NF-based AD methods. Moreover, our HGAD also surpasses the SOTA unified AD methods, PMAD (by 3.9% and 2.3%) and UniAD (by 1.9% and 1.1%), demonstrating our superiority. Furthermore, the results on BTAD, MVTec3D-RGB, and VisA (see Tab. 2) also verify the superiority of our method, where we outperform UniAD by 1.2%, 6.4%, and 4.3% in anomaly detection. The results on the Union dataset further show that our method is more superior when there are more product classes.

**Qualitative Results.** Fig. 4 shows qualitative results. It can be found that our approach can generate much better anomaly score maps than the single-class NF-based baseline CFLOW (Gudovskiy et al., 2022) even for different anomaly types. More qualitative results are in the App. Fig. 6.

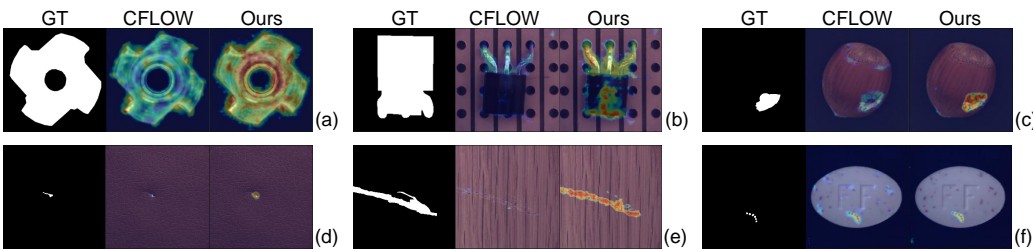

Figure 4: **Qualitative results on MVTecAD**. (a) and (b) both represent global anomalies, (c) contains large cracks, (d) shows small dints, (e) contains texture scratches, and (f) shows color anomalies.

Table 3: **Ablation studies on MVTecAD**. *SGC*, *FMC*, *ICG*, *MIM*, and *Intra* mean single Gaussian center, fixed multiple centers, inter-class Gaussian mixture prior, mutual information maximization, and intra-class mixed class centers learning, respectively. Note that the experiments in (a), (b), (c), and (e) are conducted with $\lambda_1$ and $\lambda_2$ set to 1 and 10.

(a) Hierarchical Gaussian mixture prior.

| SGC | FMC | **ICG** | **MIM** | **Intra** | Det. | Loc. |
|---|---|---|---|---|---|---|
| ✓ | - | - | - | - | 89.0 | 94.0 |
| - | ✓ | - | - | - | 94.7 | 96.1 |
| - | - | ✓ | - | - | 94.5 | 96.7 |
| - | - | ✓ | ✓ | - | 96.3 | 96.6 |
| - | ✓ | - | - | ✓ | 96.3 | 97.1 |
| - | - | ✓ | ✓ | ✓ | **97.7** | **97.6** |

(b) Number of intra-class centers.

| # Centers | Det. | Loc. |
|---|---|---|
| 3 | 97.5 | 97.3 |
| 5 | 97.6 | 97.1 |
| **10** | **97.7** | **97.6** |
| 15 | 97.6 | 97.3 |
| 20 | 97.4 | 97.2 |

(c) Anomaly criterion.

| **Logps** | **Entropy** | Det. | Loc. |
|---|---|---|---|
| ✓ | - | 94.4 | 97.1 |
| - | ✓ | 96.6 | 97.0 |
| ✓ | ✓ | **97.7** | **97.6** |

(d) Hyperparameters.

| $\lambda_1$ | $\lambda_2$ | Det. | Loc. | $\lambda_1$ | $\lambda_2$ | Det. | Loc. |
|---|---|---|---|---|---|---|---|
| 1 | 1 | 96.5 | 96.7 | 0.5 | 100 | 98.4 | 97.8 |
| 1 | 5 | 97.6 | 97.3 | **1** | 100 | **98.4** | **97.9** |
| 1 | 10 | 97.7 | 97.6 | 5 | 100 | 98.3 | 97.9 |
| 1 | 50 | 98.2 | 97.8 | 10 | 100 | 98.3 | 97.8 |
| **1** | **100** | **98.4** | **97.9** | 20 | 100 | 97.8 | 97.6 |

(e) Optimization strategy.

| | Det. | Loc. |
|---|---|---|
| - | 96.5 | 97.0 |
| ✓ | **97.7** | **97.6** |

## 4.3 ABLATION STUDIES

**Hierarchical Gaussian mixture Prior.** 1) Tab. 3a verifies our confirmation that multiple latent class centers are of vital significance. With the fixed multiple centers (FMC), image-level and pixel-level AUROCs can be improved by 5.7% and 2.1%, respectively. By employing the learnable inter-class Gaussian mxiture prior to model the latent multi-class distribution, the pixel-level AUROC can be improved by 0.6%. 2) The effectiveness of Mutual information maximization (MIM) is proven in Tab. 3a, where adding MIM brings promotion by 1.8% for detection. This shows that to better learn the complex multi-class distribution, it is necessary to endow the model class discrimination ability to avoid multiple centers collapsing into the same center. 3) Tab. 3a confirms the efficacy of intra-class mixed class centers learning. With the FMC as the baseline, introducing to learn intra-class mixed class centers could bring an increase of 1.6% for detection and 1.0% for localization, respectively. Finally, combining these, we form the hierarchical Gaussian mixture to achieve the best results.

**Number of Intra-Class Centers.** We conduct experiments to investigate the influence of intra-class centers in each class. The results are shown in Tab. 3b. The best performance is achieved with a moderate number: 10 class centers. A larger class center number like 20 does not bring further promotion, which may be because the class centers are saturated and more class centers are harder to train. For other datasets, we also use 10 as the number of intra-class centers.

**Anomaly Criterion.** Only taking the log-likelihood and the entropy as the anomaly criterion can achieve a good performance, while our associated criterion outperforms each criterion consistently. This illustrates that the associated anomaly scoring strategy is more conducive to guarantee that anomalies are recognized as out-of-distribution.

**Hyperparameters.** We ablate the hyperparameters $\lambda_1$ and $\lambda_2$ in Tab. 3d. The results in Tab. 3d show that the larger $\lambda_2$ can achieve better unified AD performance. The larger $\lambda_2$ can urge the network more focused on separating different class features into their corresponding class centers, indicating that the class discrimination ability is of vital significance to accomplish multi-class anomaly detection.

## 5 CONCLUSION

In this paper, we focus on how to unify anomaly detection regarding multiple classes. For such a challenging task, popular normalizing flow based AD methods may fall into a "homogeneous mapping" issue. To address this, we propose a novel HGAD against learning the bias with three key improvements: inter-class Gaussian mixture prior, mutual information maximization, and intra-class mixed class centers learning strategy. Under the multi-class AD setting, our method can improve NF-based AD methods by a large margin, and also surpass the SOTA unified AD methods.

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

APPENDIX

## A  MORE DISCUSSIONS

### A.1  CLASS LABELS

Our method uses class labels, but it does not introduce any extra data collection cost compared to one-for-one AD models. Because our method only explicitly uses class labels, while they implicitly use class labels. The existing AD datasets are collected for one-for-one anomaly detection (*i.e.*, we need to train a model for each class). Thus, the existing AD datasets need to be separated according to classes, with each class as a subdataset. Therefore, one-for-one AD methods also need class labels, as they require normal samples from the same class to train. If the classes are not separated (in other words, without class labels), these one-for-one AD methods cannot be used either. Our method actually has the same supervision as these methods. The difference is that we explicitly use the class labels but they don't explicitly use the class labels. So, our method still follows the same data organization format as the one-for-one AD models, we don't introduce any extra data collection cost. But the advantage of our unified AD method is that we can train one model for all classes, greatly reducing the resource costs of training and deploying. Moreover, our method only uses class labels and does not require pixel-level annotations. For real-world industrial applications, this doesn't incur extra data collection costs, as we usually consciously collect data according to different classes. So, class labels are bonuses for us.

We summarize the training samples and the supervision information required by our method and other methods as follows:

Table 4: Training samples and supervision information summarization.

| PaDiM | MKD | DRAEM | PMAD | UniAD | OmniAL | FastFlow | CFLOW | HGAD (Ours) |
|-------|-----|-------|------|-------|--------|----------|-------|-------------|
| N     | N   | N+P   | N    | N     | N+P    | N        | N     | N           |
| C     | C   | C     | C    | w/o C | w/o C  | C        | C     | C           |

where N means only using normal samples during training, P means also using pseudo (or synthetic) anomalies during training, C means requiring class labels and w/o C means not using class labels.

Our method can be easily extended to completely unsupervised, as industrial images often have significant differences between different classes. For instance, after extracting global features, we can use a simple unsupervised clustering algorithm to divide each image into a specific class. Or we can only require few-shot samples for each class as a reference, and then compute the feature distances between each input sample to these reference samples. In this way, we can also conveniently divide each sample into the most relevant class.

### A.2  MORE DISCUSSIONS WITH UNIAD

Here, we provide more discussions between our HGAD and UniAD (You et al., 2022) to further clarify that we does not simply replace the reconstruction model of UniAD with normalizing flows and accordingly introduce the "homogeneous mapping" issue.

Our method and UniAD (You et al., 2022) both aim to tackle the multi-class AD task, but we adopt a different research line from UniAD. Unlike UniAD, our method doesn't face the "identical shortcut" issue, as it doesn't have the abnormal information leakage risk in principle (see the second paragraph in sec. 1). However, the NF-based models have their own problems when used for multi-class anomaly detection. We first empirically observe a significant performance degradation when directly using the previous NF-based AD methods for multi-class anomaly detection (see Fig. 2(a) and Tab. 1). The "homogeneous mapping" issue is a possible explanation we provide for this phenomenon, rather than casually introduced. Moreover, as analyzed in sec. 3.2, we provide a reasonable explanation from the perspective of the formula in normalizing flow. Finally, the key designs proposed in our method are completely different from those in UniAD. Based on these designs, we can achieve much better unified AD performance than UniAD on all four real-world AD datasets. Therefore, compared to UniAD, our method should be reasonably seen as an effective exploration for multi-class anomaly detection in the direction of normalizing flow based anomaly detection.

Table 5: **Complexity comparison** between our HGAD and other baseline methods.

| | PaDiM | MKD | DRAEM | PMAD | UniAD | FastFlow | CFLOW | HGAD (Ours) |
|---|---|---|---|---|---|---|---|---|
| FLOPs | 14.9G | 24.1G | 198.7G | 52G | 9.7G | 36.2G | 30.7G | 32.8G |
| Learnable Parameters | / | 24.9M | 97.4M | 163.4M | 9.4M | 69.8M | 24.7M | 30.8M |
| Inference Speed | 12.8fps | 23fps | 22fps | 10.8fps | 29fps | 42.7fps | 24.6fps | 24.3fps |
| Training Epochs | / | 50 | 700 | 300 | 1000 | 400 | 200 | 100 |

## A.3 THE WAY TO GUARANTEE ANOMALIES OUT-OF-DISTRIBUTION.

Here, we further explain how to guarantee that anomalies are out-of-distribution. In our method, increasing inter-class distances is to ensure that the latent space has sufficient capacity to accommodate the features of multiple classes. In addition, we also model the intra-class Gaussian mixture distribution for each class to ensure that the normal distribution of each class still remains compact. Therefore, even if anomalies fall into the inter-class Gaussian mixture distribution, they are usually in the low-density regions among the inter-class class centers. So, we can still ensure that anomalies are out-of-distribution through intra-class Gaussian mixture distributions. As described in Anomaly Scoring section (sec. 3.4), we can guarantee that anomalies are recognized as out-of-distribution by combining intra-class log-likelihood and inter-class entropy to measure anomalies. Because only if the anomaly is out-of-distribution, the anomaly score based on the association of log-likelihood and entropy will be high, and the detection metrics can be better. The visualization results (decision-level results based on log-likelihood) in Fig. 2 and 5 also intuitively show that our method has fewer normal-abnormal overlaps and the normal boundary is more compact.

## A.4 LIMITATIONS

In this paper, we propose a novel HGAD to accomplish the multi-class anomaly detection task. Even if our method manifests good unified AD performance, there are still some limitations of our work. Here, we discuss two main limitations as follows:

One limitation is that our method mainly targets NF-based AD methods to improve their unified AD abilities. To this end, our method cannot be directly utilized to the other types of anomaly detection methods, such as reconstruction-based, OCC-based, embedding-based, and distillation-based approaches (see Related Work, Sec. 2). However, we believe that the other types of anomaly detection methods can also be improved into unified AD methods, but we need to find and solve the corresponding issues in the improvement processes, such as the "identical shortcut" issue (You et al., 2022) in reconstruction-based AD methods. How to upgrade the other types of anomaly detection methods to unified AD methods and how to find a general approach for unified anomaly detection modeling will be the future works.

In this work, our method is mainly aimed at solving multi-class anomaly detection, it doesn't have the ability to directly generalize to unseen classes. Because, in our method, the new class features usually do not match the learned known multi-class feature distribution, which can lead to normal samples being misrecognized as anomalies. Generalization to unseen classes can be defined as cross-class anomaly detection (Yao et al., 2023b), where the model is trained with normal instances from multiple known classes with the objective to detect anomalies from unseen classes. In the practical industrial scenarios, models with cross-class anomaly detection capabilities are very valuable and necessary, because new products will continuously appear and it's cost-ineffective and inconvenient to retrain models for new products. We think our method should achieve better performance on unseen classes than previous NF-based methods due to the ability to learn more complex multi-class distribution, but it's far from solving the cross-class problem. How to design a general approach for cross-class anomaly detection modeling will be the future works.

## A.5 MODEL COMPLEXITY

With the image size fixed as $256 \times 256$, we compare the FLOPs and learnable parameters with all competitors. In Tab. 5, we can conclude that the advantage of HGAD does not come from a larger model capacity. Compared to UniAD, our method requires fewer epochs (100 vs. 1000) and has a shorter training time.

## A.6 REAL-WORLD APPLICATIONS

In industrial inspection scenarios, the class actually means a type of product on the production line. Multi-class anomaly detection can be applied to train one model to detect defects in all products, without the need to train one model for each type of product. This can greatly reduce the resource costs of training and deploying. In video surveillance scenarios, we can use one model to simultaneously detect anomalies in multiple camera scenes.

## B    SOCIAL IMPACTS AND ETHICS

As a unified model for multi-class anomaly detection, the proposed method does not suffer from particular ethical concerns or negative social impacts. All datasets used are public. All qualitative visualizations are based on industrial product images, which doesn't infringe personal privacy.

## C    IMPLEMENTATION DETAILS

**Optimization Strategy.** In the initial a few epochs, we only optimize with $\mathcal{L}_g$ and $\mathcal{L}_{mi}$ to form distinguishable inter-class main class centers. And then we simultaneously optimize the intra-class delta vectors and the main class centers with the overall loss $\mathcal{L}$ in Eq. 11. In this way, we can better decouple the inter-class and intra-class learning processes. This strategy can make the intra-class learning become much easier, as optimizing after forming distinguishable inter-class main centers will not have the problem that many centers initially overlap with each other.

**Model Architecture.** The normalizing flow model in our method is mainly based on Real-NVP (Dinh et al., 2017) architecture, but the convolutional subnetwork in Real-NVP is replaced with a two-layer MLP network. As in Real-NVP, the normalizing flow in our model is composed of the so-called coupling layers. All coupling layers have the same architecture, and each coupling layer is designed to tractably achieve the forward or reverse affine coupling transformation (Dinh et al., 2017) (see Eq. 4). Then each coupling layer is followed by a random and fixed soft permutation of channels Ardizzone et al. (2019) and a fixed scaling by a constant, similar to ActNorm layers introduced by (Kingma & Dhariwal, 2019). For the coupling coefficients (*i.e.*, $\exp(s(x_1))$ and $t(x_1)$ in Eq. 4), each subnetwork predicts multiplicative and additive components simultaneously, as done by (Dinh et al., 2017). Furthermore, we adopt the soft clamping of multiplication coefficients used by (Dinh et al., 2017). The layer numbers of the normalizing flow models are all 12. We add positional embeddings to each coupling layer, which are concatenated with the first half of the input features (*i.e.*, $x_1$ in Eq. 4). Then, the concatenated embeddings are sent into the subnetwork for predicting couping coefficients. The dimension of all positional embeddings is set to 256. The implementation of the normalizing flows in our model is based on the FrEIA library `https://github.com/VLLHD/FrEIA`.

## D    DATASETS

**MVTecAD.** The MVTecAD (Bergmann et al., 2019) dataset is widely used as a standard benchmark for evaluating unsupervised image anomaly detection methods. This dataset contains 5354 high-resolution images (3629 images for training and 1725 images for testing) of 15 different product categories. 5 classes consist of textures and the other 10 classes contain objects. A total of 73 different defect types are presented and almost 1900 defective regions are manually annotated in this dataset.

**BTAD.** The BeanTech Anomaly Detection dataset (Mishra et al., 2021) is an another popular benchmark, which contains 2830 real-world images of 3 industrial products. Product 1, 2, and 3 of this dataset contain 400, 1000, and 399 training images respectively.

**MVTecAD-3D.** The MVTecAD-3D (Bergmann et al., 2021) dataset is recently proposed for 3D anomaly detection, which contains 4147 high-resolution 3D point cloud scans paired with 2D RGB images from 10 real-world categories. In this dataset, most anomalies can also be detected only through RGB images. Since we focus on image anomaly detection, we only use RGB images of the MVTecAD-3D dataset. We refer to this subset as MVTec3D-RGB.

**VisA.** The Visual Anomaly dataset (zou et al., 2022) is a recently proposed larger anomaly detection dataset compared to MVTecAD (Bergmann et al., 2019). This dataset contains 10821 images with

9621 normal and 1200 anomalous samples. In addition to images that only contain single instance, the VisA dataset also have images that contain multiple instances. Moreover, some product categories of the VisA dataset, such as Cashew, Chewing gum, Fryum and Pipe fryum, have objects that are roughly aligned. These characteristics make the VisA dataset more challenging than the MVTecAD dataset, whose images only have single instance and are better aligned.

## E   DETAILED LOSS FUNCTION DERIVATION

In this section, we provide the detailed derivation of the loss functions proposed in the main text, including $\mathcal{L}_g$ (Eq. 6), $\mathcal{L}_{mi}$ (Eq. 8), and $\mathcal{L}_{in}$ (Eq. 10).

**Derivation of $\mathcal{L}_g$.** We use a Gaussian mixture model with class-dependent means $\mu_y$ and unit covariance $\mathbb{I}$ as the inter-class Gaussian mixture prior, which is defined as follows:

$$p_Z(z) = \sum_y p(y)\mathcal{N}(z; \mu_y, \mathbb{I}) \tag{12}$$

Below, we use $c_y$ as a shorthand of $\log p(y)$. Then, we can calculate the log-likelihood as follows:

$$\begin{aligned}
\log p_Z(z) &= \log\Big[\sum_y p(y)\mathcal{N}(z; \mu_y, \mathbb{I})\Big] \\
&= \log\Big[\sum_y p(y)(2\pi)^{-\frac{d}{2}}\mathrm{e}^{-\frac{1}{2}(z-\mu_y)^T(z-\mu_y)}\Big] \\
&= -\frac{d}{2}\log(2\pi) + \log\Big(\sum_y \mathrm{e}^{c_y} \cdot \mathrm{e}^{-\frac{||z-\mu_y||_2^2}{2}}\Big) \\
&= -\frac{d}{2}\log(2\pi) + \log\Big(\sum_y \mathrm{e}^{-\frac{||z-\mu_y||_2^2}{2}+c_y}\Big) \\
&= -\frac{d}{2}\log(2\pi) + \operatorname*{logsumexp}_y\left(-\frac{||z-\mu_y||_2^2}{2} + c_y\right)
\end{aligned} \tag{13}$$

Then, we bring the $\log p_Z(z)$ into Eq. 1 to obtain the log-likelihood $\log p_\theta(x)$ as:

$$\log p_\theta(x) = -\frac{d}{2}\log(2\pi) + \operatorname*{logsumexp}_y\left(-\frac{||\varphi_\theta(x)-\mu_y||_2^2}{2} + c_y\right) + \log|\det J| \tag{14}$$

Further, the maximum likelihood loss in Eq. 2 can be written as:

$$\begin{aligned}
\mathcal{L}_m &= \mathbb{E}_{x \sim p(X)}[-\log p_\theta(x)] \\
&= \mathbb{E}_{x \sim p(X)}\left[-\operatorname*{logsumexp}_y\left(-\frac{||\varphi_\theta(x)-\mu_y||_2^2}{2} + c_y\right) - \log|\det J| + \frac{d}{2}\log(2\pi)\right]
\end{aligned} \tag{15}$$

The loss function $\mathcal{L}_g$ is actually defined as the above maximum likelihood loss $\mathcal{L}_m$ with inter-class Gaussian mixture prior.

**Extending $\mathcal{L}_g$ for Learning Intra-Class Mixed Class Centers.** When we extend the Gaussian prior $p(Z|y) = \mathcal{N}(\mu_y, \mathbb{I})$ to mixture Gaussian prior $p(Z|y) = \sum_{i=1}^M p_i(y)\mathcal{N}(\mu_i^y, \mathbb{I})$, where $M$ is the number of intra-class latent centers, the likelihood of latent feature $z$ can be calculated as follows:

$$p_Z(z) = \sum_y p(y)\Big(\sum_{i=1}^M p_i(y)\mathcal{N}(\mu_i^y, \mathbb{I})\Big) \tag{16}$$

Then, following the derivation in Eq. 13, we have:

$$\log p_Z(z) = \log\Big(\sum_y p(y)\operatorname*{sumexp}_i\Big[\frac{-||z-\mu_i^y||_2^2}{2} + c_i^y - \frac{d}{2}\log(2\pi)\Big]\Big) \tag{17}$$

where $c_i^y$ is the shorthand of $\log p_i(y)$. The $\mathcal{L}_g$ for learning intra-class mixed class centers can be defined as:

$$\mathcal{L}_g = \mathbb{E}_{x \sim p(X)}\left[-\log\Big(\sum_y p(y)\operatorname*{sumexp}_i\Big[\frac{-||\varphi_\theta(x)-\mu_i^y||_2^2}{2} + c_i^y - \frac{d}{2}\log(2\pi)\Big]\Big) - \log|\det J|\right] \tag{18}$$

However, as the initial latent features $Z$ usually have large distances with the intra-class centers $\{\mu_i^y\}_{i=1}^M$, this will cause the value after sumexp operation close to 0. After calculating the logarithm function, it's easy to cause the loss to be numerically ill-defined (NaN). Besides, we find that directly employing Eq. 18 for learning intra-class mixed class centers will lead to much worse results, as we need to simultaneously optimize all intra-class centers of all classes to fit the inter-class Gaussian mixture prior. To this end, we propose to decouple the inter-class Gaussian mixture prior fitting and the intra-class latent centers learning. The loss function of learning intra-class mixed class centers is defined in Eq. 22.

**Derivation of $\mathcal{L}_{mi}$.** We first derive the general format of the mutual information loss in Eq. 7 as follows:

$$
\begin{aligned}
\mathcal{L}_{mi} &= -I(Y, Z) = -H(Y) + H(Y|Z) = -H(Y) - H(Z) + H(Y, Z) \\
&= -H(Y) - \mathbb{E}_{x \sim p(X)} \Big[ -\log\big( \sum_y p(y)p(\varphi_\theta(x)|y) \big) \Big] \\
&\quad + \mathbb{E}_{(x,y) \sim p(X,Y)}[-\log(p(y)p(\varphi_\theta(x)|y))] \\
&= -H(Y) - \mathbb{E}_{(x,y) \sim p(X,Y)} \left[ \log \frac{p(y)p(\varphi_\theta(x)|y)}{\sum_{y'} p(y')p(\varphi_\theta(x)|y')} \right] \\
&= -\mathbb{E}_{y \sim p(Y)}[-\log p(y)] - \mathbb{E}_{(x,y) \sim p(X,Y)} \left[ \log \frac{p(y)p(\varphi_\theta(x)|y)}{\sum_{y'} p(y')p(\varphi_\theta(x)|y')} \right]
\end{aligned}
\tag{19}
$$

Then, by replacing $p(\varphi_\theta(x)|y)$ with $\mathcal{N}(\varphi_\theta(x); \mu_y, \mathbb{I})$ in the mutual information loss, we can derive the following practical loss format for the second part of Eq. 19. We also use $c_y$ as a shorthand of $\log p(y)$.

$$
\begin{aligned}
&- \mathbb{E}_{(x,y) \sim p(X,Y)} \left[ \log \frac{p(y)p(\varphi_\theta(x)|y)}{\sum_{y'} p(y')p(\varphi_\theta(x)|y')} \right] \\
&= -\mathbb{E}_{(x,y) \sim p(X,Y)} \left[ \log \frac{p(y)\mathcal{N}(\varphi_\theta(x); \mu_y, \mathbb{I})}{\sum_{y'} p(y')\mathcal{N}(\varphi_\theta(x); \mu_{y'}, \mathbb{I})} \right] \\
&= -\mathbb{E}_{(x,y) \sim p(X,Y)} \left[ \log \frac{(2\pi)^{-\frac{d}{2}} e^{-\frac{1}{2}(\varphi_\theta(x) - \mu_y)^T (\varphi_\theta(x) - \mu_y)} \cdot e^{c_y}}{\sum_{y'} (2\pi)^{-\frac{d}{2}} e^{-\frac{1}{2}(\varphi_\theta(x) - \mu_{y'})^T (\varphi_\theta(x) - \mu_{y'})} \cdot e^{c_{y'}}} \right] \\
&= -\mathbb{E}_{(x,y) \sim p(X,Y)} \left[ \log \frac{e^{-\frac{||\varphi_\theta(x) - \mu_y||_2^2}{2} + c_y}}{\sum_{y'} e^{-\frac{||\varphi_\theta(x) - \mu_{y'}||_2^2}{2} + c_{y'}}} \right] \\
&= -\mathbb{E}_{(x,y) \sim p(X,Y)} \left[ \operatorname*{logsoftmax}_y \left( -\frac{||\varphi_\theta(x) - \mu_{y'}||_2^2}{2} + c_{y'} \right) \right]
\end{aligned}
\tag{20}
$$

By replacing Eq. 20 back to the Eq. 19, we can obtain the following practical loss format of the mutual information loss.

$$
\begin{aligned}
\mathcal{L}_{mi} &= -\mathbb{E}_{y \sim p(Y)}[-\log p(y)] - \mathbb{E}_{(x,y) \sim p(X,Y)} \left[ \operatorname*{logsoftmax}_y \left( -\frac{||\varphi_\theta(x) - \mu_{y'}||_2^2}{2} + c_{y'} \right) \right] \\
&= -\mathbb{E}_{y \sim p(Y)}[-c_y] - \mathbb{E}_{(x,y) \sim p(X,Y)} \left[ \operatorname*{logsoftmax}_y \left( -\frac{||\varphi_\theta(x) - \mu_{y'}||_2^2}{2} + c_{y'} \right) \right] \\
&= -\mathbb{E}_{(x,y) \sim p(X,Y)} \left[ \operatorname*{logsoftmax}_y \left( -\frac{||\varphi_\theta(x) - \mu_{y'}||_2^2}{2} + c_{y'} \right) - c_y \right]
\end{aligned}
\tag{21}
$$

**Intra-Class Mixed Class Centers Learning Loss.** The loss function for learning the intra-class class centers is actually the same as the $\mathcal{L}_g$ in Eq. 6. But we note that we need to replace the class centers with the intra-class centers: $\mu_i^y = \{\mu_1^y + \Delta\mu_i^y\}_{i=1}^M$, and the sum operation is performed on all intra-class centers $\mu_i^y$ within the corresponding class $y$. Another difference is that we need to detach the main center $\mu_1^y$ from the gradient graph and only optimize the delta vectors. The loss function can be written as:

$$
\mathcal{L}_{in} = \mathbb{E}_{(x,y) \sim p(X,Y)} \left[ -\operatorname*{logsumexp}_i \left( -\frac{||\varphi_\theta(x) - (SG[\mu_1^y] + \Delta\mu_i^y)||_2^2}{2} + c_i^y \right) - \log|\det J| \right]
\tag{22}
$$

Finally, we note that the use of $\mathrm{logsumexp}$ and $\mathrm{logsoftmax}$ pytorch operations above is quite important. As the initial $||\varphi_\theta(x) - \mu_y||_2^2/2$ distance values are usually large, if we explicitly perform the $\exp$ and then $\log$ operations, the values will become too large and the loss will be numerically ill-defined (NaN).

## F  AN INFORMATION-THEORETIC VIEW

Information theory (Shannon, 1948) is an important theoretical foundation for explaining deep learning methods. The well-known *Information Bottleneck principle* (Tishby et al., 1999; Tishby & Zaslavsky, 2015; Alemi et al., 2017; Saxe et al., 2018) is also rooted from the information theory, which provides an explanation for representation learning as the trade-off between information compression and informativeness retention. Below, we denote the input variable as $X$, the latent variable as $Z$, and the class variable as $Y$. Formally, in this theory, supervised deep learning attempts to minimize the mutual information $I(X, Z)$ between the input $X$ and the latent variable $Z$ while maximizing the mutual information $I(Z, Y)$ between Z and the class $Y$:

$$\min I(X, Z) - \alpha I(Z, Y) \tag{23}$$

where the hyperparameter $\alpha > 0$ controls the trade-off between compression (*i.e.*, redundant information) and retention (*i.e.*, classification accuracy).

In this section, we will show that our method can be explained by the *Information Bottleneck principle* with the learning objective $\min I(X, Z_\mathcal{E}) - \alpha I(Z, Y)$, where $Z_\mathcal{E} = \varphi_\theta(X + \mathcal{E})$ and $p(\mathcal{E}) = \mathcal{N}(0, \sigma^2 \mathbb{I})$ is Gaussian with mean zero and covariance $\sigma^2 \mathbb{I}$. First, we derive $I(X, Z_\mathcal{E})$ as follows:

$$I(X, Z_\mathcal{E}) = I(Z_\mathcal{E}, X) = H(Z_\mathcal{E}) - H(Z_\mathcal{E}|X)$$
$$= \underbrace{\mathbb{E}_{x \sim p(X), \epsilon \sim p(\mathcal{E})}[-\mathrm{log} p(\varphi_\theta(x + \epsilon))]}_{:=A} + \underbrace{\mathbb{E}_{x \sim p(X), \epsilon \sim p(\mathcal{E})}[\mathrm{log} p(\varphi_\theta(x + \epsilon)|x)]}_{:=B} \tag{24}$$

To approximate the second item $(B)$, we can replace the condition $x$ with $\varphi_\theta(x)$, because $\varphi_\theta$ is bijective and both conditions convey the same information (Ardizzone et al., 2020).

$$B = \mathbb{E}_{x \sim p(X), \epsilon \sim p(\mathcal{E})}[\mathrm{log} p(\varphi_\theta(x + \epsilon)|x)] = \mathbb{E}_{x \sim p(X), \epsilon \sim p(\mathcal{E})}[\mathrm{log} p(\varphi_\theta(x + \epsilon)|\varphi_\theta(x))] \tag{25}$$

We can linearize $\varphi_\theta(x + \epsilon)$ by its first order Taylor expansion: $\varphi_\theta(x + \epsilon) = \varphi_\theta(x) + J\epsilon + \mathcal{O}(\epsilon^2)$, where the matrix $J = \bigtriangledown_x \varphi_\theta(x)$ is the Jacobian matrix of the bijective transformation ($z = \varphi_\theta(x)$ and $x = \varphi_\theta^{-1}(z)$). Then, we have:

$$B = \mathbb{E}_{x \sim p(X), \epsilon \sim p(\mathcal{E})}[\mathrm{log} p(\varphi_\theta(x) + J\epsilon + \mathcal{O}(\epsilon^2)|\varphi_\theta(x))]$$
$$= \mathbb{E}_{x \sim p(X), \epsilon \sim p(\mathcal{E})}[\mathrm{log} p(\varphi_\theta(x) + J\epsilon|\varphi_\theta(x))] + \mathbb{E}_{\epsilon \sim p(\mathcal{E})}[\mathcal{O}(\epsilon^2)]$$
$$= \mathbb{E}_{x \sim p(X), \epsilon \sim p(\mathcal{E})}[\mathrm{log} p(\varphi_\theta(x) + J\epsilon|\varphi_\theta(x))] + \mathcal{O}(\sigma^2) \tag{26}$$

where the $\mathbb{E}_{\epsilon \sim p(\mathcal{E})}[\mathcal{O}(\epsilon^2)]$ is actually the covariance of $p(\mathcal{E}) = \mathcal{N}(0, \sigma^2 \mathbb{I})$, thus can be replaced with $\mathcal{O}(\sigma^2)$. Since $p(\mathcal{E})$ is Gaussian with mean zero and covariance $\sigma^2 \mathbb{I}$, the conditional distribution is Gaussian with mean $\varphi_\theta(x)$ and covariance $\sigma^2 JJ^T$. Then, we have:

$$B = \mathbb{E}_{x \sim p(X), \epsilon \sim p(\mathcal{E})}[\mathrm{log} \mathcal{N}(\varphi_\theta(x) + J\epsilon; \varphi_\theta(x), \sigma^2 JJ^T)] + \mathcal{O}(\sigma^2)$$
$$= \mathbb{E}_{x \sim p(X), \epsilon \sim p(\mathcal{E})}[\mathrm{log}((2\pi)^{-\frac{d}{2}} \cdot (|\sigma^2 JJ^T|)^{-\frac{1}{2}} \cdot \mathrm{e}^{-\frac{1}{2}\frac{1}{\sigma^2}\epsilon^T\epsilon})] + \mathcal{O}(\sigma^2)$$
$$= \mathbb{E}_{x \sim p(X)}[-\frac{1}{2}\mathrm{log}(|\sigma^2 JJ^T|)] - \frac{d}{2}\mathrm{log}(2\pi) - \frac{1}{2\sigma^2}\mathbb{E}_{\epsilon \sim p(\mathcal{E})}[\epsilon^T\epsilon] + \mathcal{O}(\sigma^2)$$
$$= \mathbb{E}_{x \sim p(X)}[-\frac{1}{2}\mathrm{log}(|JJ^T|)] - d\mathrm{log}(\sigma) - \frac{d}{2}\mathrm{log}(2\pi) - \frac{1}{2\sigma^2}\mathcal{O}(\sigma^2) + \mathcal{O}(\sigma^2)$$
$$= \mathbb{E}_{x \sim p(X)}[-\mathrm{log}|\mathrm{det} J|] - d\mathrm{log}(\sigma) - \frac{d}{2}\mathrm{log}(2\pi) - \frac{1}{2\sigma^2}\mathcal{O}(\sigma^2) + \mathcal{O}(\sigma^2) \tag{27}$$

For the first item (A), we can use the derivation in Eq. 13.

$$
\begin{aligned}
A &= \mathbb{E}_{x \sim p(X), \epsilon \sim p(\mathcal{E})}[-\log p(\varphi_\theta(x + \epsilon))] \\
&= \mathbb{E}_{x \sim p(X), \epsilon \sim p(\mathcal{E})}\Big[\frac{d}{2}\log(2\pi) - \underset{y}{\text{logsumexp}}\Big(-\frac{||\varphi_\theta(x + \epsilon) - \mu_y||_2^2}{2} + c_y\Big)\Big] \\
&= \mathbb{E}_{x \sim p(X), \epsilon \sim p(\mathcal{E})}\Big[-\underset{y}{\text{logsumexp}}\Big(-\frac{||\varphi_\theta(x + \epsilon) - \mu_y||_2^2}{2} + c_y\Big)\Big] + \frac{d}{2}\log(2\pi) \quad (28)
\end{aligned}
$$

Finally, we put the above derivations together and drop the constant items and the items that vanish with rate $\mathcal{O}(\sigma^2)$ as $\sigma \to 0$. The $I(X, Z_\mathcal{E})$ becomes:

$$
I(X, Z_\mathcal{E}) = \mathbb{E}_{x \sim p(X), \epsilon \sim p(\mathcal{E})}\Big[-\underset{y}{\text{logsumexp}}\Big(-\frac{||\varphi_\theta(x + \epsilon) - \mu_y||_2^2}{2} + c_y\Big) - \log|\det J|\Big] \quad (29)
$$

We can find that the $I(X, Z_\mathcal{E})$ has the same formula as the loss $\mathcal{L}_g$ except the constant item $\frac{d}{2}\log(2\pi)$, and $I(Z, Y) = I(Y, Z) = -\mathcal{L}_{mi}$ (see Eq. 19). Thus, the learning objective $\min I(X, Z_\mathcal{E}) - \alpha I(Z, Y)$ in *Information Bottleneck principle* can be converted to $\mathcal{L}_g + \alpha \mathcal{L}_{mi}$, which is the first half part of the training loss in Eq. 11.

From the *Information Bottleneck principle* perspective, we can explain our method: it attempts to minimize the mutual information $I(X, Z_\mathcal{E})$ between $X$ and $Z_\mathcal{E}$, forcing the model to ignore the irrelevant aspects of $X + \mathcal{E}$ which do not contribute to fit the latent distribution and only increase the potential for overfitting. Therefore, the $\mathcal{L}_g$ loss function actually endows the normalizing flow model with the compression ability for establishing correct invertible mappings between input $X$ and the latent Gaussian mixture prior $Z$, which is effective to prevent the model from learning the "homogeneous mapping". Simultaneously, it encourages to maximize the mutual information $I(Y, Z)$ between $Y$ and $Z$, forcing the model to map different class features to their corresponding class centers which can contribute to class discriminative ability.

## G    ADDITIONAL RESULTS

**Quantitative Results Under the Single-Class Setting.** In Tab. 6, we report the detailed results of anomaly detection and localization on MVTecAD (Bergmann et al., 2019) under the single-class setting. We can find that all baselines achieve excellent results under the single-class setting, but their performances drop dramatically under the unified case (see Tab. 1 in the main text). For instance, the strong baseline, DRAEM, suffers from a drop of 9.9% and 10.1%. The performance of the previous SOTA NF-based AD method, FastFlow, drops by 7.6% and 2.5%. This demonstrates that the unified anomaly detection is quite more challenging than the conventional single-class anomaly detection task, and current SOTA AD methods cannot be directly applied to the multi-class AD task well. Thus, how to improve the unified AD ability for AD methods should be further studied. On the other hand, compared with reconstruction-based AD methods (*e.g*, DRAEM (Zavrtanik et al., 2021)), NF-based AD methods have less performance degradation when directly applied to the unified case, indicating that NF-based approaches may be a more suitable way for the multi-class AD modeling than the reconstruction-based approaches.

**Log-likelihood Histograms.** In Fig. 5, we show log-likelihoods generated by the single-class NF-based AD method and our method. All categories are from the MVTecAD dataset. The visualization results can empirically verify our speculation that the single-class NF-based AD methods may fall into the "homogeneous mapping" issue, where the normal and abnormal log-likelihoods are highly overlapped.

**Qualitative Results.** We present in Fig. 6 additional anomaly localization results of categories with different anomalies in the MVTecAD dataset. It can be found that our approach can generate much better anomaly score maps that the single-class NF-based baseline CFLOW (Gudovskiy et al., 2022) even for different categories from the MVTecAD dataset.

Table 6: **Anomaly detection and localization results on MVTecAD**. All methods are evaluated under the single-class setting. ·/· means the image-level and pixel-level AUROCs.

| Category | Baseline Methods | | | Unified Methods | | NF Based Methods | |
| | PaDiM | MKD | DRAEM | PMAD | UniAD | FastFlow | CFLOW |
|---|---|---|---|---|---|---|---|
| Carpet | 99.8/99.0 | 79.3/95.6 | 97.0/95.5 | 99.7/98.8 | 99.9/98.0 | 100/99.4 | 100/99.3 |
| Grid | 96.7/97.1 | 78.0/91.8 | 99.9/99.7 | 97.7/96.3 | 98.5/94.6 | 99.7/98.3 | 97.6/99.0 |
| Leather | 100/99.0 | 95.1/98.1 | 100/98.6 | 100/99.2 | 100/98.3 | 100/99.5 | 97.7/99.7 |
| Tile | 98.1/94.1 | 91.6/82.8 | 99.6/99.2 | 100/94.4 | 99.0/91.8 | 100/96.3 | 98.7/98.0 |
| Wood | 99.2/94.1 | 94.3/84.8 | 99.1/96.4 | 98.0/93.3 | 97.9/93.4 | 100/97.0 | 99.6/96.7 |
| Bottle | 99.9/98.2 | 99.4/96.3 | 99.2/99.1 | 100/98.4 | 100/98.1 | 100/97.7 | 100/99.0 |
| Cable | 92.7/96.7 | 89.2/82.4 | 91.8/94.7 | 98.0/97.5 | 97.6/96.8 | 100/98.4 | 100/97.6 |
| Capsule | 91.3/98.6 | 80.5/95.9 | 98.5/94.3 | 89.8/98.6 | 85.3/97.9 | 100/99.1 | 99.3/99.0 |
| Hazelnut | 92.0/98.1 | 98.4/94.6 | 100/99.7 | 100/98.8 | 99.9/98.8 | 100/99.1 | 96.8/98.9 |
| Metal nut | 98.7/97.3 | 73.6/86.4 | 98.7/99.5 | 99.2/97.5 | 99.0/95.7 | 100/98.5 | 91.9/98.6 |
| Pill | 93.3/95.7 | 82.7/89.6 | 98.9/97.6 | 94.3/95.5 | 88.3/95.1 | 99.4/99.2 | 99.9/99.0 |
| Screw | 85.8/98.4 | 83.3/96.0 | 93.9/97.6 | 73.9/91.4 | 91.9/97.4 | 97.8/99.4 | 99.7/98.9 |
| Toothbrush | 96.1/98.8 | 92.2/96.1 | 100/98.1 | 91.4/98.2 | 95.0/97.8 | 94.4/98.9 | 95.2/99.0 |
| Transistor | 97.4/97.6 | 85.6/76.5 | 93.1/90.9 | 99.8/97.8 | 100/98.7 | 99.8/97.3 | 99.1/98.0 |
| Zipper | 90.3/98.4 | 93.2/93.9 | 100/98.8 | 99.5/96.7 | 96.7/96.0 | 99.5/98.7 | 98.5/99.1 |
| **Mean** | 95.5/97.4 | 87.8/90.7 | 98.0/97.3 | 96.1/96.8 | 96.6/96.6 | 99.4/98.5 | 98.3/98.6 |

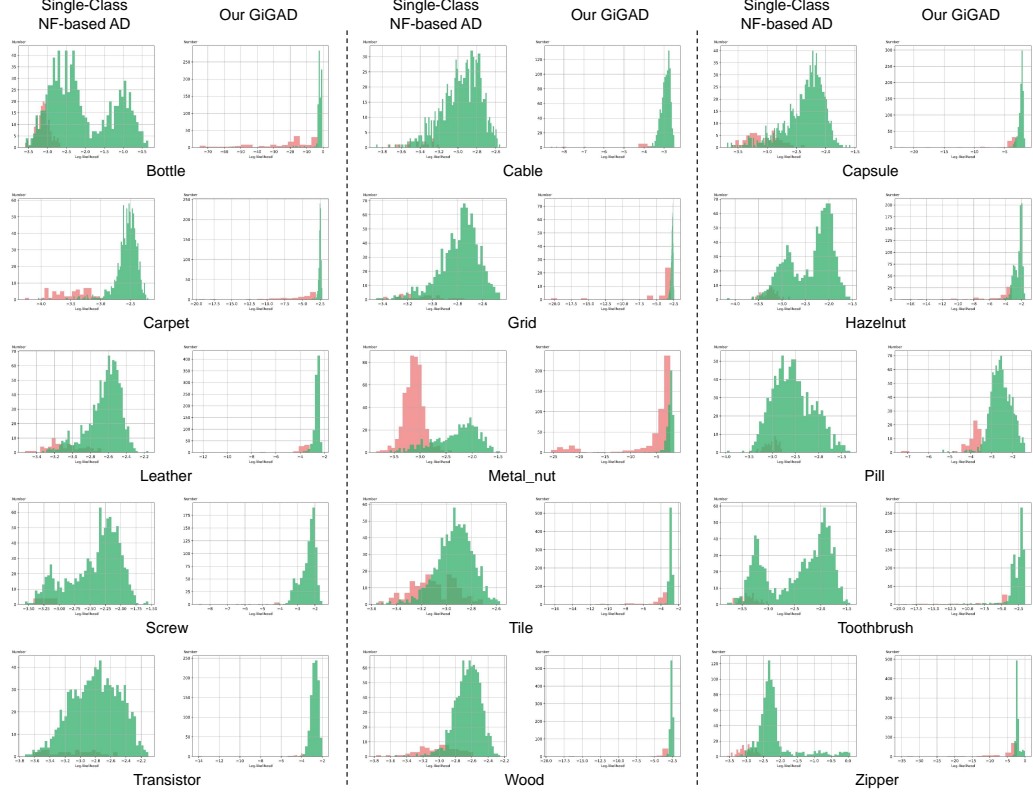

Figure 5: **Log-likelihood histograms on MVTecAD**. All categories are from the MVTecAD dataset.

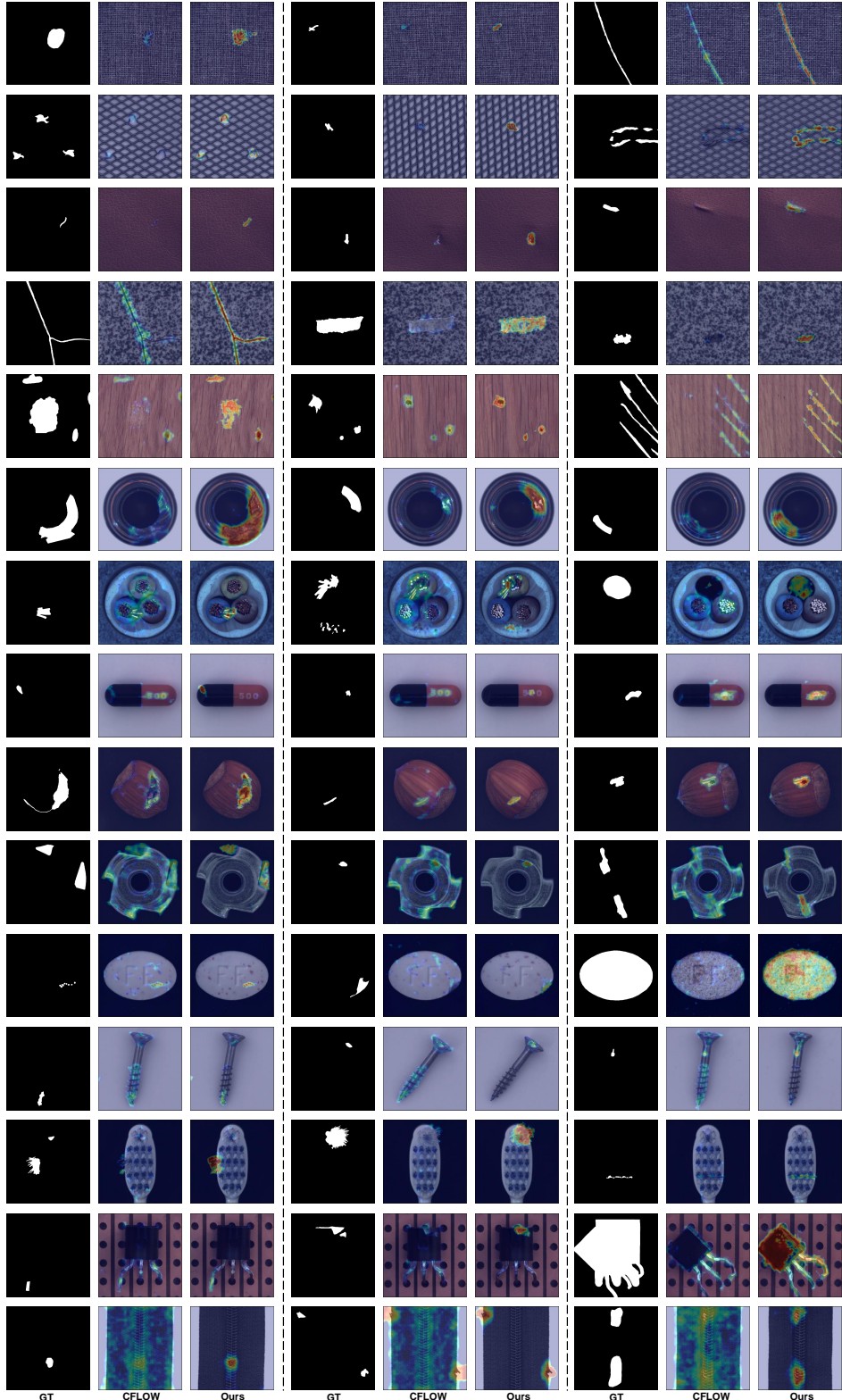

Figure 6: **Qualitative results on MVTecAD**. More visualization of anomaly localization maps generated by our method on industrial inspection data. All examples are from the MVTecAD dataset.

