# OpenReview forum: "Hierarchical Gaussian Mixture Normalizing Flows Modeling for Multi-Class Anomaly Detection"
_ICLR.cc/2024/Conference — Submitted to ICLR 2024_

### Official Review · Reviewer_v6Sb · 2023-10-27

**Soundness:** 2 fair
**Presentation:** 1 poor
**Contribution:** 2 fair
**Rating:** 3
**Confidence:** 3

**Summary:**

The authors proposed a normalizing flow-based unified anomaly detection method, i.e., Hierarchical Gaussian Anomaly Detection (HGAD). By designing a loss function, the proposed method attempts to handle the intra-class diversity and inter-class separation.

**Strengths:**

* The proposed method can model normalizing flows with hierarchical Gaussian mixture priors on the multi-class anomaly detection task.
* The design of the proposed method can model the inter-class Gaussian mixture priors, learn class centers, and maximize mutual information.

**Weaknesses:**

* The presentation and the layout of the manuscript are bad. For examples,
  * Figure 2(b)/(d): The authors need to clarify the colors associated with the classes.
  * What are the $\lambda_1, \lambda_2$ in Section 4.2? It is confusing that the authors list several separate loss functions in Section 3.3 without any articulation about how to deal with these equations to achieve the goal(s). I finally found the objective function of the target goal after checking with the appendix, but the authors didn't mention anything in the main paper.
  * In Section 3.4: What did the authors mean by level-k? Additionally, since there is no access to the label in the test, which $y$ in $\mu_i^y$ will be used for the test point?
  * The limited explanation between problem formulation and the experiment setup:
    * In Section 3.2, since the authors pointed out that Eq. (2) is used to maximize the log-likelihood of normal features, why do the normalizing flows present a large log-likelihood for both normal and abnormal features? In other words, are both normal and abnormal/anomaly observations used in this loss function?
    * In Section 4, what is the partition for the data in experiments? What are the normal classes? What are the anomaly classes? If label information (including anomalies) is used in the training, why do we call this multi-class anomaly detection? What is the difference between this with the regular multi-class classification?

* Since there are multiple goals contained in the objective function and different training strategies in the experiments, to clearly summarize the work, it would be better to use pseudocode to outline the algorithm.

* The weak support of the necessity of the intra-class centers: From Figure 3(b), I cannot see there is a significant difference among different numbers of intra-centers.

**Questions:**

* Section 3.1: Why $p_\theta$ is a probability rather than a density? If it is a density function, why did the authors subtract that from 1 (any motivation)?

* Figure 3: Why is the positional encoding added to the normalizing flow? Is this necessary? Did the authors conduct the ablation study of this design?

* Bottom in Page 5: Why do not just use sample class priors to estimate $p(Y)$? Which part of the architecture in Figure 3 is used to estimate $p(Y)$? Could the authors explain in detail?

* The notations in (8), and (9) are bad. What is $\mu\_{y^\prime}$? Do you mean the center vector? Is loss (9) necessary? Why there is no penalty cost before this loss in the final object function? Did the authors conduct the ablation study for this loss function?

* The discussion in Section 3.3:
  * Could the authors further clarify this sentence: "Because our method only explicitly uses class labels, while they implicitly use class labels (see App. A)".
  * I see one loss function is designed to maximize the log-likelihood of normal observations. Why did the author claim that using a label should not be a strict condition? Did the author conduct the experiment to support this conclusion?

**Details Of Ethics Concerns:**

No.

---

> ### Author Response · Authors · 2023-11-13
>
> **[To W1].** The red means abnormal and the green means normal. In anomaly detection papers, authors usually use red to represent abnormal and green for normal, we are a bit careless in not clearly indicating the corresponding classes of the two colors in Figure 2. We will add a legend to the figure to indicate the meaning of two colors in the revision.
>
> **[To W2].** We are sorry about this confusion. Due to page limitation, we move the overall loss function to the Appendix. This results in the absence of definitions of hyperparameters $\lambda_1$ and $\lambda_2$. We really appreciate that you point out this problem. We will revise the layout of our paper to ensure that such a problem won't appear in our paper.
>
> **[To W3].** The Feature Extractor will extract multi-scale feature maps, level-k means the kth feature map level. Using multi-scale features is common in anomaly detection, and our method also builds a normalizing flow model at each feature level. We appreciate your suggestion and will add explanations for level-k in the revision.
>
> Firstly, we will explain the difference between multi-class anomaly detection (AD) and multi-class classification, you may misunderstand our multi-class AD task. Multi-class classification focuses on classifying input samples, while anomaly detection focuses on detecting abnormal regions in input samples. Previous AD methods follow the one-for-one paradigm (i.e., we train one model for each class). In multi-class AD, we only train one unified model for all classes. Thus, $y$ doesn't indicate whether the input is normal or abnormal but indicates what class the input belongs to. Unlike multi-class classification, we do not know whether the input is normal or abnormal, but we can know which class it belongs to. Please also see our responses to W5 and Q5.
>
> **[To W4].** In anomaly detection, we only use normal samples for training as our goal is to detect anomalies. In our paper, we have clearly explained that normalizing flow (NF) may have a ”homogeneous mapping'' problem when used for multi-class anomaly detection, where NF may take a bias to map different input features to similar latent features (i.e., this means anomalies will have similar log-likelihoods to normal). Moreover, the goal of training is to maximize the log-likelihoods of normal features. Thus, normalizing flows will represent large likelihoods for both normal and abnormal features. Please see the sec. 3.2 in our paper, we have provided detailed explanations for this.
>
> **[To W5].** Firstly, we will indicate the difference between industrial anomaly detection and semantic anomaly detection. Our paper focuses on industrial anomaly detection rather than semantic anomaly detection. In industrial anomaly detection, all classes are normal, the anomalies are defective areas that exist in the image and don't have classes. Only in semantic anomaly detection (e.g., CIFAR10, CIFAR100, and ImageNet-30 are usually the datasets), we will select some classes as normal and the other classes are used as abnormal.
>
> The partition follows the standard way of these industrial AD datasets, MVTecAD, BTAD, MVTec3D-RGB, and VisA. Generally speaking, most industrial AD papers will not specifically elaborate on the data partition, as the training and test sets are fixed.
>
> We have clearly defined the multi-class AD in the second paragraph of Introduction: one unified model is trained with normal samples from multiple classes, and the objective is to detect anomalies in these classes. In training, we only use normal samples from multiple product classes without any anomaly, the label information is used to indicate one normal sample belongs to which class. For the last question, please see our response to W3.
>
> **[To W6].** Thanks for your suggestion, we will add the pseudocodes of our algorithm in the revision.
>
> **[To W7].** We validate the effectiveness of the intra-class centers in ablation studies (see Tab. 3(a)). The number of intra-class centers is the same for each class (we state this in our paper, please see sec. 4.2, Setup). Figure 3 is just a schematic diagram of our model. Moreover, we can see that the number of intra-class centers in the two classes is the same, but the intra-class distributions are different (this is our purpose of introducing intra-class centers).

---

> ### Author Response · Authors · 2023-11-13
>
> **[To Q1].** $p_\theta(x) = {\rm e}^{{\rm log}p_\theta(x)}$ is coming from the log-likelihood ${\rm log}p_\theta(x)$ by exponential function. We call it probability as the value of $p_\theta(x)$ is in $(0,1)$. Our statement is: $s(x) = 1 - p_\theta(x)$, $p_\theta(x)$ is a value for representing normality, $s(x)$ is thus a value for representing abnormality.
>
> **[To Q2].** It's necessary. With positional embeddings, we can achieve better results. We don't conduct an ablation study for this design, as the effectiveness of positional embeddings is already validated in the previous work CFLOW [1]. In our paper, we also state that the normalizing flow used in our model is the same as the one in CFLOW (please see sec. 4.2, Quantitative Results).
>
> [1] Denis Gudovskiy, Shun Ishizaka, and Kazuki Kozuka. Cflow-ad: Real-time unsupervised anomaly
> detection with localization via conditional normalizing flows. In IEEE Winter Conference on
> Application of Computer Vision, 2022.
>
> **[To Q3].** Because we think that parameterization $p(Y)$ can enable the network to adaptively learn the class weights, and parameterization $p(Y)$ only introduces a small number of parameters. The parameter to control $p(Y)$ is $\psi$, and $\psi$ is learned by optimizing the E.q(6). Then $p(y)$ is estimated by ${\rm softmax}_y(\psi)$. No module in Figure 3 is dedicated to estimating $p(Y)$, the parameter $\psi$ belongs to the normalizing flow model.
>
> **[To Q4].** We respectfully disagree with this comment. $\mu_{y^\prime}$ means all other class centers except the class center $\mu_y$ corresponding to $y$. Because our method is used for multi-class anomaly detection, there will be many class centers $\{\mu_y\}, y \in \{1,\dots,N\}$. In Eq.(8) and (9), when we employ the softmax function to calculate the value for a special class $y$, all the other classes are naturally represented as $y^\prime$. We think that such notations are commonly used in the softmax function. So, we don't specifically explain $y^\prime$. We will add corresponding explanations to make E.q (8) and (9) easier to understand in the revision.
>
> Loss (9) is necessary. In Tab 3(c), we show that using Entropy is beneficial to achieve better results. And loss (9) is used to optimize entropy during training. The total loss is $\mathcal{L} = \lambda _1\mathcal{L} _g + \lambda _2\mathcal{L} _{mi} + \mathcal{L} _e + \mathcal{L} _{in}$. Our design is to use $ \mathcal{L} _e $ and $ \mathcal{L} _{in} $ as auxiliary losses. As there are four optimization objectives, adding a weighting factor and conducting ablation studies for each loss item will result in a large number of combinational experiments, which will bring us too much burden.
>
> **[To Q5].** In Appendix A, we have clearly explained this statement. The existing AD datasets are collected for one-for-one anomaly detection (i.e., we need to train a model for each class). Thus, the existing AD datasets need to be separated according to classes, with each class as a subdataset. Therefore, one-for-one AD methods also need class labels, as they require normal samples from the same class to train, but they don't explicitly use the class label. Our method can achieve only training a unified model for all classes, this is a significant innovation compared to the one-for-one AD methods. Moreover, our method still follows the same data organization format as the one-for-one AD methods, but we need to explicitly assign a label for each class. This actually doesn't introduce any extra data collection cost. Please see Appendix A.1 for more discussions.
>
> **[To Q6].** As we mentioned above, the label is used to distinguish which class each sample belongs to. In a stricter condition, we may hope the multi-class AD models even don't need labels, which means that we have many normal samples from different classes but don't know which class they belong to. Because the AD datasets themselves separate samples according to classes, we can easily get the class label from these datasets. Therefore, we claim that using class labels should not be a strict condition (or constraint) when designing multi-class AD methods. We think that we explain this clearly, this conclusion seems not to require experimental validation.

---

### Official Review · Reviewer_tw2T · 2023-10-31

**Soundness:** 3 good
**Presentation:** 2 fair
**Contribution:** 3 good
**Rating:** 6
**Confidence:** 3

**Summary:**

This paper proposes a normalizing flow model with hierarchical Gaussian mixture prior for unified anomaly detection, HGAD. This method achieves the SOTA unified AD performance on four datasets.

**Strengths:**

1. The analysis and discussion of the proposed model are detailed.

2. The experiment results are superior to the comparison methods.

3. The experiments in both the formal paper and appendix are relatively thorough.

**Weaknesses:**

1. The abstraction is somewhat lengthy. Please polish the abstraction and make it concise.

2. The size of coordinate/legend in Figure 2 is too small to recognize.

3. The representation should be improved to be more professional. The explanations of some equations (eg. Eq6 and Eq9 ) are not easy-understood.

**Questions:**

1. Is the homogeneous mapping issue intrinsically equal to the well-known identical shortcut problem?

2. The citations might be wrong. Many citations should be placed in the brackets. Please pay attention to the difference between '\citep' and '\citet'.

3. The full name of HGAD should be listed.

4. Why the performance of multi-class case is lower than the unified case, as shown in Table 1.

5. The best performance on MVTec in Table 1 are 98.4/97.9, but 97.7/97.6 in Table 3.

---

> ### Author Response · Authors · 2023-11-13
>
> **[To W1].** We are very grateful for this suggestion, we will polish the abstract to make it more concise in the reavision.
>
> **[To W2].** Thanks for your suggestion. We will attempt to use a larger font size to plot the figure for making the coordinate/legend large enough to directly read.
>
> **[To W3].** Thanks for your suggestion. E.q(6) is a derived loss (the detailed derivation is in Appendix E), we will include some key intermediate steps for assisting understanding. E.q(9) is actually the entropy formula, where we use $-||\varphi_\theta(x)-\mu_y||^2_2/2$ as class logits. $\mu_{y^\prime}$ means all the other class centers except for $\mu_y$. We don't specifically explain $\mu_{y^\prime}$ as we think this notation is commonly used in the softmax function. We will add more explanations to make E.q(9) easier to understand. For other equations, we will also carefully check and explain these equations more clearly in the revision.
>
> **[To Q1].** Thank you for the comment, but we cannot fully agree with this comment. The identical shortcut is essentially caused by the leakage of abnormal information. The process of reconstruction is to remove abnormal information in the input, resulting in the failure of reconstruction in abnormal regions. But if the reconstruction network is overfitted, the abnormal features may be leaked into the output, resulting in the reconstruction network directly returning a copy of the input as output. So, the reconstruction errors in abnormal regions will be small, leading to abnormal missing detection. This issue usually can be addressed by masking, such as the neighbor masking mechanism in UniAD.
>
> Homogeneous mapping is a specific issue in normalizing flow (NF) based AD methods. In previous NF-based AD methods, the latent feature space has a single center. When used for multi-class AD, we need to map different class features to the single latent center, this may cause the model more prone to take a bias to map different input features to similar latent features. Thus, with the bias, the log-likelihoods of abnormal features will become closer to the log-likelihoods of normal features, causing normal misdetection or abnormal missing detection. To address this issue, we propose hierarchical Gaussian Mixture normalizing flow modeling. Because there are significant differences in the causes and solutions of the two issues, we think that the two issues are not intrinsically equal. In Appendix A.2, we have provided a thorough discussion for this question.
>
> **[To Q2].** We are very grateful for this comment, it's really helpful for us to improve our paper. We will revise this issue in the revision.
>
> **[To Q3].** HGAD is taken from our paper title: **H**ierarchical **G**aussian Mixture Normalizing Flows Modeling for Multi-class **A**nomaly **D**etection. As this is too long, we didn't list the full name in the main text (only list “Hierarchical Gaussian mixture” in Abstract). In the revision, we will explain the HGAD naming in the main text.
>
> **[To Q4].** In our paper, multi-class and unified actually have the same meaning. In the caption of Tab 1, we use “unified/multi-class” to express that multi-class is an alias for unified. The meaning of results is explained by the following sentence: “$\cdot$/$\cdot$ means the image-level and pixel-level AUROCs''. We are sorry for this misunderstanding. We will revise this misleading caption in the revision.
>
> **[To Q5].** In Tab 3(a), the hyperparameters $\lambda_1$ and $\lambda_2$ are set as 1 and 10. However, we later find that setting $\lambda_1$ and $\lambda_2$ to 1 and 100 can get better results (see Tab 3(d)). In Tab 1, we update the better results, but most results in Tab 3 are obtained with the $\lambda_1$ and $\lambda_2$ set as 1 and 10.

---

> > ### Comment · Reviewer_tw2T · 2023-11-22
> >
> > Thanks for your response. However, the explanation about Q1 is not convincing to me.
> > The overall representations need to be improved in the revised version. (Note that ICLR allows authors to upload the modified versions.)

---

> > > ### Author Response · Authors · 2023-11-24
> > >
> > > Thanks for your response. Could you please further tell us the reason why the explanation about Q1 is not convincing to you? We may be able to further discuss this question to give both sides a better understanding. Thanks for your reminder. We have carefully revised our paper based on the suggestions of all reviewers these days, but the submission link was closed when we were to upload the revised paper. We are sorry for being currently unable to upload our revised paper. We would like to thank you again for your constructive suggestions, we believe that the revised paper has made good improvements compared to the previous one.

---

> > > > ### Author Response · Authors · 2023-11-29
> > > > **Modified version**
> > > >
> > > > We found that the submission link was opened again, and thus we have uploaded the revised paper. Please feel free to see the revised paper.

---

### Official Review · Reviewer_37tJ · 2023-11-23

**Soundness:** 3 good
**Presentation:** 2 fair
**Contribution:** 2 fair
**Rating:** 5
**Confidence:** 3

**Summary:**

The paper addresses the problem of *supervised* *multiclass* anomaly detection, where the "normal" samples may belong to a pre-defined set of classes Y, and the goal is to detect anomalous samples that do not belong to any class in Y. The authors point to drawbacks with prior reconstruction-based and normalizing-flow (NF) based approaches to multiclass anomaly detection. They then propose a new approach for alleviating these drawbacks by building on existing NF-based methods replacing their unimodal Gaussian prior with a hierarchical Gaussian mixture prior. Experimental results and ablation studies demonstrate that the proposed approach is better on average compared to prior methods.

**Strengths:**

- Extending existing NF-based approaches with a mixture of Gaussian prior looks like a natural approach to take for multiclass anomaly detection
- Fairly extensive experimentation with ablation studies that attempt to show the role of individual loss components

**Weaknesses:**

- One of my main concerns is that most of the methods compared to (e.g. UniAD, FastFlow, etc) are *unsupervised* whereas the proposed method is a *supervised* approach explicitly requiring class labels to be provided (see e.g. discussion in Appendix A.1). On the face of it, this does not seem like a fair comparison to make. It is important that the authors explicitly summarize what supervision each method uses and justify why theirs is a better approach despite requiring explicit label information to be provided during training.

- The writing and presentation is at places hard to follow. The authors are urged to present the high-level approach first before dwelling into the details of the individual loss components. Having an explicit pseudo-code stating what the supervision is for the algorithm, and how the overall optimization objective looks like would be very helpful.

- The proposed approach appears to have a lot of moving parts: there are four loss components (one for a inter-class Gaussian mixture, one for an intra-class Gaussian mixture, a mutual information based and an entropy-based loss for class diversity), with two hyper-parameters for weighting them (Appendix C). Although the authors do conduct some analysis of different hyper-parameter combinations, one if left with a feeling that the approach is highly heuristic in nature, with the gains coming largely from heavy engineering effort. Improving the writing and presentation may help boost the reader's confidence in the proposed method.

**Questions:**

Of the methods discussed, it appears that BGAD is supervised, but not compared to. Are there other methods you compare to in experiments which like your method are also supervised?

---

> ### Author Response · Authors · 2023-11-23
>
> **[To W1].** In the anomaly detection field, we think that our method should not be seen as supervised. Because anomaly detection focuses on distinguishing anomalies from normals, methods that use abnormal samples during training should be seen as supervised. For example, BGAD is supervised as it uses few-shot anomalies during training. However, our method only uses normal samples during training.
>
> As we have explained in Appendix A.1, our method has the same data organization format as the one-for-one AD methods (e.g., PaDiM, MKD, DRAEM, FastFlow, and CFLOW in our paper) and doesn't introduce any extra data collection cost. The existing AD datasets are collected for one-for-one anomaly detection (i.e., we need to train a model for each class). Thus, the existing AD datasets need to be separated according to classes, with each class as a subdataset. Therefore, one-for-one AD methods also need class labels, as they require normal samples from the same class to train. If the classes are not separated (in other words, without class labels), these one-for-one AD methods cannot be used either. Our method actually has the same supervision as these methods. The difference is that we explicitly use the class labels but they don't explicitly use the class labels. However, our advantage is that we can achieve only training a unified model for all classes, this is a significant innovation compared to the one-for-one AD methods. The unified AD method, PMAD, also explicitly uses class labels when accomplishing multi-class anomaly detection. The UniAD and OmniAL don't use class labels, but as we mentioned above, using class labels doesn't introduce any extra data collection cost. However, our method can achieve better multi-class detection results than UniAD and OmniAL.
>
> In a stricter condition, we may hope the multi-class AD models even don’t need labels, which means that we have many normal samples from different classes but don’t know which class
> they belong to. Because the AD datasets themselves separate samples according to classes, we can easily get the class labels from these datasets. Therefore, we think that using class labels should not be a strict condition (or constraint) when designing multi-class AD methods.
>
> **[To W2].** Thanks for your suggestion. In the revision, we will provide an explicit pseudo-code of our algorithm to more clearly describe the whole algorithm process and the overall optimization object to make readers more easier to follow our paper.
>
> **[To W3].** Thanks for this comment, it’s really helpful for us to improve our paper. We will take your suggestions and other reviewers' suggestions seriously to improve the presentation and organization order of our paper. We would be very grateful if you could provide us with more suggestions on further improving our paper.
>
> **[To Q1].** As we discussed in **W1**, BGAD uses few-shot anomalies during training while our method doesn't. The data requirements of our method and BGAD are quite different, so it should be reasonable that we don't compare with BGAD. Moreover, we have compared with the SOTA NF-based AD methods, FastFlow and CFLOW, and this has demonstrated the effectiveness of our method. Our method has the same data organization format as the one-for-one AD methods, such as PaDiM, MKD, DRAEM, FastFlow, and CFLOW. The unified AD method, PMAD, also explicitly uses class labels. From the data requirement perspective, these methods are like our method, but we can achieve multi-class anomaly detection or better multi-class AD results.

---

> > ### Comment · Reviewer_37tJ · 2023-11-28
> > **Thank you for the clarification**
> >
> > I thank the authors for the detailed response. This is very helpful.
> >
> > I think the paper would benefit from a summary of what supervision each method receives. I agree that my referencing your method as "supervised" is not entirely accurate, given that you don't get to see anomalies samples.
> >
> > From your response, it appears that UniAD and FastFlow do not need the class labels to be specified. Despite this, they appear to be close competitors (especially OmniAL). While there are certainly applications where procuring labeled data is easy, I can also imagine scenarios where obtaining a large corpus on unlabeled "valid" samples is a lot cheaper than getting labeled data.
> >
> > I would like to discuss this paper further with the other reviewers and AC before finalizing the score.

---

> > > ### Author Response · Authors · 2023-11-29
> > > **Thanks for your response**
> > >
> > > Thanks for your response. Anomaly detection papers usually call methods that only use normal samples for training as unsupervised, strictly using supervised or unsupervised to distinguish among the methods in our paper may be ambiguous. But we can indicate the training samples and the supervision information required by these methods as follows:
> > >
> > > |PaDiM | MKD | DRAEM | PMAD | UniAD | OmniAL | FastFlow | CFLOW | Ours |
> > > | --- | --- | --- | --- | --- | --- | --- | --- | --- |
> > > |  N | N | N+P | N | N | N+P | N | N | N |
> > > |  C | C | C | C | w/o C | w/o C | C | C | C|
> > >
> > > where N means only using normal samples during training, P means also using pseudo (or synthetic) anomalies during training, C means using class labels and w/o C means not using class labels.
> > >
> > > If we think that using synthetic anomalies introduces anomalous information during training, DRAEM and OmniAL can also be called as supervised or self-supervised, while others are unsupervised. In addition, both UniAD and OmniAL use additional information to simulate anomalies. UniAD adds noise while OmniAL uses synthetic anomalies to learn how to reconstruct anomalies into normal during training. But our method is entirely based on learning normal feature distribution without any additional information (If synthetic anomalies can be used, our method can easily be combined with BGAD to achieve better multi-class AD results). However, the methods based on synthetic anomalies may perform much worse when synthetic anomalies cannot simulate real anomalies well. This will result in limited application scenarios for such methods. For example, on the more challenging VisA dataset, our method significantly outperforms OmniAL (97.1/98.9 vs. 87.8/96.6). A note: we report 94.2/96.0 for OmniAL in the paper, but we reviewed the OmniAL paper and found that 94.2/96.0 is under the separate case and 87.8/96.6 is the result under the unified case. We will correct this in the revision. Compared to UniAD, results on multiple datasets, such as MVTec3D-RGB, VisA, and Union datasets, also show significant improvements (83.9 vs. 77.5, 97.1 vs. 92.8, 92.3 vs. 86.9).

---

### Official Review · Reviewer_EGfe · 2023-12-12

**Soundness:** 2 fair
**Presentation:** 2 fair
**Contribution:** 3 good
**Rating:** 3
**Confidence:** 3

**Summary:**

The paper addresses the problem of multi-class anomaly detection using normalizing flow-based generative modeling. They address the issue of "homogeneous mapping" where latent representations corresponding to both normal and anomalous inputs could map to the same isotropic Gaussian distribution. This hinders the ability of the model to detect anomalous inputs. They propose to model the latent distribution using a Gaussian mixture, with one or multiple Gaussian components corresponding to each class, which improves the flexibility of the model. Further, they design specific loss functions to ensure that the latent Gaussian distributions corresponding to distinct classes are well separated.

**Strengths:**

1. Addresses an important limitation of normalizing flow-based anomaly detection methods which typically assume that the latent distribution is the standard Gaussian. Here, they motivate and explore a class-conditional Gaussian mixture distribution for the latent variable, which increases the flexibility of the model and can improve the anomaly detection performance.

2. Experiments are fairly extensive and compare with multiple recent baselines.

**Weaknesses:**

1. The proposed method requires labeled training data, which is not usually available in anomaly detection problems. The method hinges on knowledge of of the number of classes so that in the latent space each class is modeled with either a single Gaussian or a mixture. Since some of the compared baselines do not have use this label supervision, the comparison might be unfair.

1. The overall training objective and its individual losses are not well motivated and connected together. It's not clear why the entropy based loss is needed in Eqn 9. What exactly is the entropy being estimated here and why is it not covered by the mutual information loss? In the final objective (Eqn 11), why are only two of the terms scaled by a hyper-parameters?

1. Several issues with the notations and writing which make it hard to follow. For instance:
    - In Algorithm 1: What is meant by $\mu_y \leftarrow \mathbf{y}$? On line 3, it is denoted as $x \in X^k$, but $X^k$ is not the set of all feature maps.
    - In Eqn 5, the Gaussian is denoted by $\mathcal{N}(\mu_y, \Sigma_y)$ and by $\mathcal{N}(z ; \mu_y, \Sigma_y)$ in the same line.
    - In Eqn 8, it is not clear what $y'$ is and therefore what $c_{y'}$ is. Same comment for Eqn 9.
    - The anomaly score function in section 3.4 is not clear. What is meant by $max(\sum_{k=1}^K P_k)$? Also, why is this form chosen for the score function?
    - Referring to the last line of page 3, why is the anomaly score defined as 1 minus the probability density (which could be greater than 1)? One could simply use the log-likelihood (or its negative) as the anomaly score.
    - In page 3, the Jacobian is defined like the gradient and it's actual expansion is never discussed in the paper. I realize that this is a standard expression for flow-based models, but it should be mentioned for clarity.
    - The need for positional encoding is not clearly explained.

**Questions:**

Please see my comments and questions under Weaknesses.

It is important to clarify in the problem setting that labeled training data are required.

A key prior work which models the latent distribution using a Gaussian mixture is not discussed. \
Semi-Supervised Learning with Normalizing Flows:
https://proceedings.mlr.press/v119/izmailov20a/izmailov20a.pdf

---

### Meta-Review · Area_Chair_a2j2 · 2023-12-06

**Metareview:**

The paper proposes an approach for multi-class anomaly detection based on normalizing flows. The key idea is to construct a Gaussian mixture model to define the flow, with suitable encouragement of diversity amongst the centers. This approach shows promising empirical results.

Reviewers had a mixed view of the paper, identifying the following strengths and weaknesses:
- good empirical results

- unclear use of supervision compared to baselines

- heuristic nature of the approach

- unclear presentation, including in the experimental setup

In the response, the authors clarified that the use of "supervision" is in the form of *class labels*, rather than *anomaly labels*. It was clarified that the baselines do not assume the existence of class labels. It was also argued that in some settings, the proposed method can significantly outperform the baselines (e.g., VisA dataset).

The AC additionally reviewed the paper. We agree with the concerns regarding the presentation and heuristic nature of the approach. The latter is a multi-scale issue: e.g., to ensure diversity in class means, it is suggested to use a mutual information *and* entropy regularizer. Given the method has multiple moving parts, it is unclear how broadly extensible the method would be.

The paper could have some ideas of interest to the community, but it is encouraged for the authors to work on incorporating the reviewers' suggestions on clarity, and also providing more motivation for the various steps of the algorith. A revised version would be suitable for a fresh round of reviews at some future venue.

Additional comments for consideration:
- the introduction mentions "product categories", which is a bit unclear in context

- since the authors consider the setting where class labels are present for training samples, it is worth commenting whether the goal is to jointly learn an anomaly detector *and* a classifier

- it could be clearer what *conceptual* advantages are offered by moving away from one-for-one anomaly detection

**Justification For Why Not Higher Score:**

Lack of clarity, heuristic nature of solution with many moving parts

**Justification For Why Not Lower Score:**

N/A

---

### Decision · Program_Chairs · 2024-01-16

Reject